

# Reanalysis of and attribution to near-surface ozone
# concentrations in Sweden during 1990-2013
**Camilla Andersson[1], Heléne Alpfjord[1], Lennart Robertson[1], Per Erik Karlsson[2]**
**and Magnuz Engardt[1]**
[1]{Swedish Meteorological and Hydrological Institute, SE-60176 Norrköping, Sweden}
[2]{Swedish Environmental Research Institute, P.O. Box 53021, SE-40014 Gothenburg,
Sweden}
Correspondence to: C. Andersson (camilla.andersson@smhi.se)
**Abstract**
We have constructed two data sets of hourly resolution reanalyzed near-surface ozone ($O_3$)
concentrations for the period 1990-2013 for Sweden. Long-term simulations from a
chemistry-transport model (CTM) covering Europe were combined with hourly ozone
concentration observations at Swedish and Norwegian background measurement sites using
data assimilation. The reanalysis data sets show improved performance than the original CTM
when compared to independent observations.
In one of the reanalyzes we included all available hourly near-surface $O_3$ observations, whilst
in the other we carefully selected time-consistent observations in order to avoid introducing
artificial trends. Based on the second reanalysis we investigated statistical aspects of the near-
surface $O_3$ concentration, focusing on the linear trend over the 24 year period. We show that
high near-surface $O_3$ concentrations are decreasing and low $O_3$ concentrations are increasing,
which is mirrored by observed improvement of many health and vegetation indices (apart
from those with a low threshold).
Using the chemistry-transport model we also conducted sensitivity simulations to quantify the
causes of the observed change, focusing on three processes: change in hemispheric
background, meteorology and anthropogenic emissions (Swedish and other European). The
rising low concentrations of near-surface $O_3$ in Sweden are caused by a combination of all



three processes, whilst the decrease in the highest $O_3$ concentrations is caused by $O_3$ precursor
emissions reductions.
While studying the relative impact of anthropogenic emissions changes, we identified
systematic differences in the modelled trend compared to observations that must be caused by
incorrect trends in the utilised emissions inventory or by too high sensitivity of our model to
emissions changes.
**1   Introduction**
Elevated concentrations of near-surface ozone ($O_3$) are a major policy concern, given their
ability to damage both vegetation (e.g. Royal Society, 2008) and human health (e.g. WHO,
2006). It is also an important greenhouse gas (e.g. IPCC, 2013). Elevated $O_3$ concentrations
are formed in the troposphere by the oxidation of volatile organic compounds (VOCs) and
carbon monoxide, driven by solar radiation in a polluted air mixture that includes nitrogen
oxides ($NO_x$). Close to combustion sources, the background $O_3$ concentration is reduced
through reactions with directly emitted nitric oxide (NO; see for example Finlayson-Pitts and
Pitts, 2000). However, further away from the source and with sufficient availability of VOCs
and the right weather conditions, these $NO_x$ emissions can lead to rises in the $O_3$
concentration. $O_3$ can be transported to regions far away from the area where it was formed
and even across continents (e.g. Akimoto, 2003; Derwent et al. 2015). Oxidized nitrogen can
also be transported to remote regions as reservoir species, such as peroxy-acetyl nitrates
(PANs). These can be a significant source of $NO_x$ and alongside naturally emitted biogenic
VOCs cause $O_3$ formation in otherwise non-polluted areas (e.g. Jacob et al., 1993).
European and North American anthropogenic emissions of $NO_x$ increased over most of the
20[th] century, but decreased strongly since the 1980s due to emission control (e.g. Lamarque et
al., 2010; Granier et al., 2011). Asian emissions have continued to rise under the same period
(Ohara et al., 2007). Jonson et al. (2006) showed that the trend in $O_3$ concentration in Europe
cannot be fully explained by changes in European precursor emissions. By inter-continental
transport the increasing precursor emissions in Asia could contribute to increasing
background levels with at least a strong impact in North America (Vestraeten et al., 2015),
whilst the trend in European background $O_3$ seasonal variation could also be affected by the
decreases in North American precursor emissions (Derwent et al., 2015). Climate also
changes over time, causing both changes to the $O_3$ forming potential, biogenic emissions of



O$_3$ precursors and deposition processes (Andersson and Engardt, 2010). Variability in climate,
such as the North Atlantic Oscillation (NAO), contributes to the variation in O$_3$ concentration
in the upper troposphere through variations both in the stratospheric contribution and in the
transport patterns (Gaudel et al., 2015). Although the stratospheric contribution to the O$_3$
concentration at the surface is generally small (3-5 ppb(v)) in Europe (Lelieveld and
Dentener, 2000), it can be a relevant contribution to near-surface O$_3$ in certain areas and time
periods (Zanis et al., 2014) and could affect the observed trend in near-surface O$_3$ (e.g. Fusco
and Logan, 2003). Despite the large number of studies of tropospheric O$_3$, a number of
challenges still remain, such as explaining the near-surface concentration trends (Monks et al.,

10  2015).

Observations in the northern mid-latitudes, either at the surface (Oltmans et al, 2006) or from
ozone-sondes and commercial aircraft (Logan et al., 2012), present the picture of increasing
tropospheric O$_3$ concentrations during the second half of the 20$^{th}$ century (Parrish et al., 2012;
Cooper et al., 2014). The strong increase in near-surface O$_3$ concentration until the late 1990s
at three widely separated North Atlantic sites, including Mace Head, seems to have peaked or
remained stationary (Simmonds et al 2004; Oltmans et al., 2006; Derwent et al., 2007). At
Pico Mountain Observatory in the Azores, a decreasing O$_3$ concentration trend was observed
during 2001-2011 which was believed to be mainly caused by decreasing precursor emissions
in North America (Kumar et al., 2013). Air masses with European origin observed at Mace
Head show a decrease in summertime peak O$_3$ concentrations and increase in wintertime,
which is believed to be connected to European NO$_x$ policy (Derwent et al., 2013). O$_3$
concentrations observed at European alpine sites and in ozone-sonde data (MOZAIC) above
European cities have decreased since 1998 with the strongest decrease in summer (Logan et
al., 2012).
Several modelling efforts have been conducted to describe the past near-surface O$_3$
concentration development (e.g. Fusco and Logan, 2003; Schultz et al., 2007; Pozolli et al.,
2011, Xing et al. 2015). Parrish et al. (2014) present past trends in tropospheric O$_3$
concentrations modelled with three chemistry-climate models and conclude that while there is
considerable qualitative agreement between the measurements and the models, there are also
substantial and consistent quantitative disagreements. These include that the models capture
only 50 % of the change observed during the last 5-6 decades and little of the observed
seasonal differences, and that the rate of the trends are badly captured. There are ways



forward to improve the description of the trends: 1) understanding the processes and
improving the model description of the physics and chemistry for processes of greatest
importance in these models, 2) improving the input data quality and 3) incorporating
observations in the model by data fusion methods to accurately represent the past statistics in
a reanalysis. The first two are important for conducting scenario calculations, whilst the last
is an option for producing mappings.
If correctly conducted, data fusion will improve the modelled estimates. If temporal and
spatial consistency is not considered, it may however introduce artificial trends. Data
assimilation, a subset to data fusion (Zhang et al., 2012), is the process by which observations
of a system are incorporated into the model state of a numerical model, in this case into the
chemistry transport model (CTM) (Kalnay, 2003; Denby and Spangl, 2010). Advanced data
assimilation schemes like the 4 dimensional variational (4dvar; e.g. Courtier et al., 1994;
Inness et al., 2013) technique utilize information provided by satellites and propagate this in
space and time from a limited number to a wide range of chemical components to provide
fields that are physically and chemically consistent with the observations. Inness et al. (2013)
performed a reanalysis of global chemical composition, including $O_3$ concentration, for 2003-
2010 using advanced data assimilation of satellite observations within the framework of the
monitoring atmospheric composition and climate (MACC) project. They demonstrated
improved $O_3$ and CO concentration profiles for the free troposphere, but biases remained for
the lower troposphere. Another global reanalysis using data assimilation of satellite data for
2005-2012, showed improved performance for chemical species (Miyazaki et al., 2015) but
for the $O_3$ concentration at the surface errors remain associated with low retrieval sensitivity
in the lower troposphere and gaps in spatial representation between the model and
observations. In order to improve surface characteristics, in situ observations of $O_3$ need to be
included in data assimilation. Another reanalysis of near-surface $O_3$ concentration in Europe
was conducted for the period 2003-2012 within the MACC project (Katragkou et al., 2015).
The reanalysis was based on the MACC global model, which consists of the European Centre
for Medium-Range Weather Forecasts' Integrated Forecast System (IFS) coupled to the
MOZART-3 CTM. In this reanalysis 4dvar data assimilation was used to incorporate in situ
measurement from the databases EMEP and Airbase. The data assimilation reduced the bias
in near-surface $O_3$ concentration in most of Europe, and it reproduced the summertime
maximum in most parts of Europe, but not the early spring peak in northern Europe. When
restricting the observations to in situ measurements in Europe, the beginning of the time



period of the reanalysis can be extended further back in time utilizing simpler data assimilation techniques than 4dvar. Variational analysis in 2 dimensions (2dvar) and the analytical counterpart optical interpolation can be used as a CPU-efficient diagnostic tools to improve modelled near-surface $O_3$ retrospectively (e.g. Alpfjord and Andersson, 2015; Robichaud and Ménard, 2014).

The MATCH (Multi-scale Atmospheric Transport and CHemistry) Sweden system (Alpfjord and Andersson, 2015) includes an operational CTM and methods for data assimilation of atmospheric concentrations in air and precipitation. The system is used for annual assessments of the near-surface $O_3$, $SO_2$, $NH_3$ and $NO_2$ background concentrations and deposition of nitrogen, sulfur and base cations in Sweden. In this study, the MATCH Sweden system is used to conduct a reanalysis of the hourly near-surface $O_3$ concentration for Sweden and Norway during the 24-year period 1990-2013 using 2dvar. We use time-consistent input data to avoid the introduction of artificial trends in the results. In an attempt to understand the trends, we perform model sensitivity analyses and apply the CTM without data assimilation. This approach brings new knowledge to explain the trends in $O_3$ concentrations found in Sweden.

The aims of this study are:

- To create a state-of-the art long-term, temporally and spatially consistent, reanalysis of hourly near-surface $O_3$ concentrations covering the geographical areas of Sweden and Norway (see Sect. 2)

- To evaluate the performance of the $O_3$ reanalysis of the MATCH Sweden system, used in the annual assessment of air quality in Sweden (see Sect. 3.1)

- To investigate trends and extreme values in near-surface $O_3$ in Sweden (see Sect. 3.2) and its implications on health and vegetation (see Sect. 3.4)

- To understand the causes of the change over time, focusing on contributions of emission change, lateral and upper boundary and meteorological variability. (see Sect. 3.3)



## 2 Method

In this study we utilize data assimilation in order to combine the respective best qualities of both a CTM and long-term measurements to map near-surface $O_3$ concentrations during a long historical time period (1990-2013). We focus our study on Sweden, but also include Norway in the data assimilation.

For the data assimilation we use the MATCH Sweden system, which is briefly explained in Sect. 2.1. Here variational analysis in two dimensions is applied, and further details are given in Sect. 2.4. Concentration fields provided by the CTM at each grid point are considered as the "first guess" (background field/prior information) of our "best estimate" of the state of the atmosphere before the introduction of observations (Kalnay, 2003). The method used for the production of the "first guess" is explained in Sect. 2.2. The selection of measurements that are included in the data assimilation is important, both to avoid artificial trends in the reanalysis data and in order to select observation sites with corresponding spatial and temporal representations as in the model. We explain our method for the selection of measurements in Sect. 2.3.

One aim of this study is to investigate trends in near-surface $O_3$ in Sweden. To understand the long-term changes in concentration we try to quantify the causes of change, through model sensitivity analyses, and applying the MATCH model without data assimilation. We investigate the respective contributions to the trends of change in European emissions by separating the impact on $O_3$ trends of changes in local emissions in Sweden, in hemispheric background concentrations (including changes to the top and lateral boundaries) and in meteorology (including changes to biogenic emissions, transport, $O_3$ forming capacity, $O_3$ deposition etc.). The method for this quantification is described in Sect. 2.5. The methods we use for evaluation are given in Sect. 2.6.

### 2.1 The MATCH Sweden system

The MATCH Sweden system is an operational system used for annual assessments of near-surface regional background concentrations in air of $O_3$, $NO_2$, $NH_3$ and $SO_2$ as well as deposition of sulfur, nitrogen and base cations over Sweden (Alpfjord and Andersson, 2015). The system includes an operational CTM (MATCH; Multi-scale Atmospheric Transport and Chemistry; Robertson et al., 1999) and methods for data assimilation (using 2dvar) of





atmospheric concentrations in air and precipitation. The yearly results from the mapping can
be found at www.smhi.se/klimatdata/miljo/atmosfarskemi.
The flow-chart in Fig. 1 describes the parts of the MATCH Sweden system that are used in
this reanalysis of near-surface $O_3$ concentrations. Explanations are provided in Sect. 2.2 to
2.4. For a description of the whole MATCH Sweden system, see e.g. Alpfjord and Andersson

6 (2015).

## 2.2  First guess – model assessment

The starting point (cf. Fig. 1) for the two-dimensional variational data assimilation of near-
surface $O_3$ is hourly fields of modelled $O_3$, produced by MATCH. The MATCH model
includes ozone- and particle-forming photo-chemistry with ~60 species (Langner et al., 1998;
Andersson et al., 2007, 2015). Part of the gas-phase chemical scheme was updated based on
Simpson et al. (2012), except for some reaction rates (following the recommendations by the
International Union of Pure and Applied Chemistry, IUPAC), and the isoprene chemistry
mechanism that was based on an adapted version of the Carter one-product mechanism
(Carter, 1996; Langner et al., 1998). A selection of compounds with different ozone forming
potentials is used to represent all hydrocarbons emitted into the atmosphere. The photolysis
rates depend on the photolytically active radiation, which is dependent on latitude, time of
day, cloud cover etc. In this study MATCH interpolates the input meteorology to a domain
covering Europe and surrounding areas with 44 km grid point spacing. MATCH uses all
meteorological model layers for vertical wind calculations, but restricts the calculations of
chemistry and transport to the lower troposphere using the vertical levels of the
meteorological model from the surface up to ca 5 km height.
MATCH is an offline model, thus, driven by meteorological data generated externally and as
such it is often a challenge to undertake long (multi-decadal) simulations due to non-
homogenous input data. Dynamical meteorological models, which provide the three-
dimensional meteorology for the offline CTMs, are constantly updated to higher resolutions
and more advanced physical schemes. Emission inventories are typically constructed for
certain target years and different methods may have been used to compile total emissions
and/or the geographical distribution of the emissions. Careless combination of different
emission data or meteorology from varying model configurations can introduce artificial





secular trends in the modelling of atmospheric pollutants. Emissions of biogenic isoprene are
calculated online in MATCH following the E-94 isoprene emission methodology proposed by
Simpson et al. (1995). Further details of MATCH in the present model version and its ability
to simulate near-surface $O_3$ can be found in separate publications, for example Markakis et al.
(2016), Lacressoniere et al. (2016) and Watson et al. (2015; 2016). In this study, we
specifically aimed for internally coherent input data, although it led to compromises in e.g.
the temporal coverage of the meteorology and the resolution of the gridded pan-European
emissions. In the following sections we briefly describe the utilized input data.

## 2.2.1 Meteorology and boundary concentrations

In the present study we force MATCH with three-dimensional meteorology from the
numerical weather forecast model HIRLAM. Within the EURO4M-project
(http://www.euro4m.eu) HIRLAM was run as forecasts from 6-hourly analyses, composed of
variational upper air analyses in 3 dimensions and optimal interpolation surface analyses.
Lateral and lower (sea surface temperature and sea ice) boundaries were taken from ERA-
Interim (Dee et al., 2011). Full three-dimensional model states needed to run MATCH are
available from 1979 through February 2014. Under EURO4M, HIRLAM was running on a
domain covering Europe and Northern Africa with 22 km grid point spacing and 60 vertical
layers from the surface to 10 hPa.
Although the present study focuses on Sweden it is necessary to realistically describe the
fluxes of $O_3$ from continental Europe and further afield. Hemispheric concentrations of all
species are similar to the ones used by Andersson et al. (2007) for the modelled year 2000. As
in Andersson et al. (2007), boundary values representative for the lateral and top boundaries
of relevant species are interpolated spatially with a monthly temporal resolution. Boundary
concentrations of $O_3$, oxidized nitrogen and methane are scaled to mimic observed changes in
the hemispheric background during the period 1990 through 2013 (cf. Fig. 2a). The same
factor is used for all months of the respective year, although most species also undergo a
seasonal cycle in the boundary concentrations used by MATCH (see supplement Fig. S1).





## 2.2.2 Emissions
The version of MATCH utilized in this study needs anthropogenic emissions of sulfur ($SO_2$
and sulfate), nitrogen oxides (NO and $NO_2$), carbon monoxide (CO), non-methane volatile
organic compounds (NMVOCs), and ammonia ($NH_3$). The model uses annually accumulated
values for each species, which are distributed with different temporal or vertical profiles
based on species and sectors.
For countries outside Sweden (as well as international shipping) we utilize the gridded (50 km
× 50 km) annual data available at EMEP's web-page (http://www.emep.int; downloaded 23
June, 2015). All emission data were split into congruent 5 km × 5 km cells where we replaced
the coarse-resolution data over Sweden with the original emission data from SMED (Svensk
miljöemissionsdata; http://www.smed.se; 1 km × 1 km converted to 5 km × 5 km cells in
EMEP's geometry). National totals from SMED are very similar to the national totals
available in the EMEP database, but our methodology enables higher resolution emission data
over Sweden. The gridded 5 km × 5 km emission data were interpolated to MATCH's 44 km
resolution domain during the simulations.
Both the total domain and Swedish anthropogenic $O_3$ precursor emissions decrease strongly
over the period 1990-2013 (cf. Fig. 2b). The total domain anthropogenic precursor emissions
decrease on average[1] by 1.8 % yr$^{-1}$, 2.4 % yr$^{-1}$, 2.6 % yr$^{-1}$ during 1990-2013 for $NO_x$,
NMVOC and CO respectively, whereas biogenic isoprene emissions (calculated online by
MATCH) increase by 0.8 % yr$^{-1}$ according to our simulations. The Swedish emissions
decrease by similar amounts (2.4 % yr$^{-1}$, 2.1 % yr$^{-1}$ and 2.9 % yr$^{-1}$). The Swedish contribution
to the total domain emissions is 1.0 % for $NO_x$ and 1.7 % for NMVOC and CO on the
average, with a slight decrease in the relative Swedish contribution over the period for $NO_x$
(0.01% yr$^{-1}$), and a slight increase for NMVOC and CO (0.01 % yr$^{-1}$ and 0.003 % yr$^{-1}$
respectively). We assume that there is no trend in the temporal intra-annual variation of the
emissions.

---

[1] The trend is calculated by linear regression over the period 1990-2013 and related to the 1990 emission level.





## 2.3   Measurements

Figures 3 and 4 summarize the observations of hourly $O_3$ concentrations used in the variational analysis and the corresponding hourly data coverage per year in the period 1990-2013. These measurements represent the regional background in Sweden and Norway. The sites included are all instrumentation sites, where $O_3$ is measured continuously and reported with hourly temporal resolution. The data assimilation is conducted on hourly resolution, which means that measurements with a coarser time resolution, such as diffusive samplers, are not included in the variational technique. Two measurement data sets were compiled (see Table 1):

- The first includes data from all available instrumentation sites in Sweden, and a selection in Norway based on data availability, quality and location. These are all the red and blue sites in Figs. 3 and 4 also including years where the data capture is lower than 80 %. The reanalysis based on these measurement data is called ALL.

-  The second data set includes data from instrumentation sites for which the data coverage exceeds 80 % for at least 23 out of the 24 years. These are the red sites in Figs. 3 and 4. The reanalysis based on these measurement data is called LONGTERM. Råö is seen as the replacement for the site Rörvik, and therefore these sites form a pair, which is included in this data set. Birkenes I was replaced by Birkenes II in 2009, and the two sites were run in parallel for a few years. We choose to include Birkenes II from 2010 and onwards. The reason for the change of site location is that Birkenes I was influenced by local effects, such as night-time inversions (personal communication with Sverre Solberg, NILU). The inclusion of these two sites could introduce an artifical trend in the reanalysis, but since it is outside the main focus area (Sweden) and mainly during night we choose to include the site in the LONGTERM reanalysis.

The two measurement data sets are input to two otherwise similar data assimilations. The ALL-reanalysis is our best estimate of gridded near-surface $O_3$ over Sweden for a given time. The LONGTERM-reanalysis is used for trend and statistical analyses. This is because changes in the number of sites and data coverage in the ALL data set can introduce artificial trends due to model biases being corrected by observations included in the later part of the period but not in the first. We return to whether these reanalyzes differ in Sect. 3.1.



## 2.4  Data assimilation

The spatial analysis problem can be formulated as how to best distribute observational
information at a discreet number of locations to a spatially consistent field. We have adopted
the 2dvar approach, which includes a modelled background field (from a CTM simulation,
"first guess") combined with available in situ observations (Robertson and Kahnert, 2007), as
indicated in Fig. 1. With this method the error estimates of both the background field and the
observations play a central role. The observational errors are assumed independent and
uncorrelated, while the background errors have spatial correlations that form a background
error matrix. The solution is found by the best combination of the background field and
observations given their respective error estimates. This can be described as a variational
problem, defined by a cost function,
$$J(x) = 0.5 \; [x\text{-}x^b]^T \; \mathbf{B}^{-1} \; [x\text{-}x^b] + 0.5 \; [y\text{-}\mathbf{H}(x)]^T \; \mathbf{O}^{-1} \; [y\text{-}\mathbf{H}(x)]$$
where x is the state to be found (the reanalysis), $x^b$ the background state (our "first guess"), y
the vector of observations, $\mathbf{H}$ is the observation operator, and $\mathbf{B}$ and $\mathbf{O}$ are the error
covariance matrices of the background field and the observations, respectively. In order to
find the optimal solution the cost function is stepwise minimized by a variational method,
starting with $x = x^b$, and ending with the state x, which represents the optimal balance between
the two terms. During the process the co-variances in the B matrix acts to extrapolate the
observational information in space.
We restrict our study to reanalyze near-surface $O_3$ on the regional background scale, which
means we only include regional background measurement sites. We also restrict our study to
2dvar, rather than using higher dimensional variational analysis. The background covariance
matrix is modelled in a simplified fashion with a constant background error, 20 times larger
than the observation error, and Gaussian spatial correlations with a length scale of 1000 km.
This implies a strong weight towards the observations and assuming a rather large horizontal
influence of the observations, which is related to the rather sparse network of regional
background observations and the relatively small emissions of $O_3$ precursors in Sweden
resulting in weak horizontal gradients of near-surface $O_3$ on the regional background scale.
The data assimilation was conducted on a 22 km resolution grid with hourly temporal
resolution, combining the modelled "first guess" for near-surface $O_3$ (the MATCH base case



scenario, MFG in Table 1) and regional background measurements. Two 24-year reanalyzes
were formed, using the two sets of hourly measurement described in Sect. 2.3 (ALL and
LONGTERM in Table 1). If an included measurement site was lacking an observation for a
specific hour, the site was excluded from the data assimilation for that specific hour.
The resulting spatially resolved hourly $O_3$ data are used to form annual and seasonal statistical
metrics for $O_3$, such as the mean value and the maximum 1-hour mean value, and annual
policy and impact related metrics (cf. Fig. 1). We analyze these annual and seasonal data for
the 1990-2013 mean, trend and extreme values in Sect. 3.2 (annual/seasonal mean and
maximum) and Sect. 3.4 (health and vegetation impact metrics).

## 2.5    Understanding the trends

We include also a quantification of the causes to the trend in near-surface $O_3$ concentration.
For this investigation we conduct model simulations with MATCH, *excluding data*
*assimilation*. We investigate the respective contributions to the modelled total trend due to

15        A. Change in emissions, which is separated between

16           o Swedish anthropogenic emissions (Se emis)

17           o European (full domain) non-Swedish anthropogenic emissions (Eur emis)

18        B. Change in lateral and upper boundaries (bound)

19        C. Change in meteorology, including online modelled biogenic isoprene emissions

20           (meteo)

Four sensitivity simulations are conducted; in which each of the four listed processes are kept
constant at the level in 2011. The respective contributions to the trend are formed by
subtracting the MFG with the corresponding sensitivity simulation. All model simulations and
scenarios are described in Table 1a. The method of forming the contributions from these
simulations is shown in Table 1b.
There are two critical points in the investigation of the causes of the trend: First, this
quantification methodology assumes linearity, whereas the sum of contributions (SUM) is not
necessarily equal to the trend in the MFG simulation. If they are not equal, this means that the
simulation is non-additive. This could occur when changes to mixtures of complex chemistry,



weather situations and emissions take place. For this reason we compare the sum of the trend in the estimated contributions to the MFG trend. Second, we quantify the contributions to the trend in the MFG simulation, which may differ from the reanalyzed trend. Thus we will compare the reanalyzed trend and the base case trend to make sure the base case simulation does not deviate too strongly from the reanalysis results. If the deviation is too large, i.e. the modelled trend is far from the observed, this means that it is non-representative. Such discrepancies could arise from over-sensitivity in MATCH to one process and insensitivity to another, compared to the real world, or imperfections/artificial trends in the input data such as erroneously estimated emissions or erroneous assumptions on the trend in hemispheric background concentrations. If either is true (non-representative or non-additive) for the trend in a specific metric, such as the trend in the January mean, then our method cannot be used to explain that specific trend.

## 2.6  Evaluation

We evaluate two aspects of the reanalysis. The first is an independent evaluation for a single year with focus on the data assimilation method. The second is an evaluation of the simulated near-surface $O_3$ concentration trend over the period and our ability to explain the causes of the trend.

For an independent evaluation of data assimilation method we conduct a cross validation at the included Swedish measurement sites. With this method we exclude one measurement site at a time from the data assimilation, and use the analysis results from the excluded location in the evaluation of all sites. This means we conduct one such 2dvar simulation for each considered measurement site. Due to the large amount of computation involved we evaluate one year only by this method. We choose the year 2013, which is when the data coverage is the largest. This means that we have the opportunity also to investigate whether we see any difference in performance between the reanalysis with the larger number of measurement sites (ALL) and the long-term reanalysis (LONGTERM). The evaluation metrics used here are mean value (mean), standard deviation ($\sigma$), model mean bias normalized by the observed mean (%bias), Pearson correlation coefficient ($r$) and the root mean square error (RMSE), see Supplement Sect. S1.





For the evaluation of the long-term trend we focus on the two critical points raised in the
previous section: 1/ the additivity of the trend in the contributions as compared to the trend in
$O_3$ concentration from the MFG simulation, and 2/ whether the MFG trend is representative
of the $O_3$ concentration trend in the LONGTERM reanalysis results. We focus this
investigation on 11 different percentiles of hourly mean $O_3$ concentrations, for an estimate of
the scores at different concentration levels. We focus specifically on averages over the three
Swedish regions North, Central and South (cf. Fig. 3), to investigate whether there is any
variation in performance in Sweden. The comparisons are presented as scatterplots in Fig. 5
and compared to the 1:1 line, factor 2 line and equal sign quadrants.
Additional evaluation and comparisons of the temporal variation over the whole period is
included in the Supplements for the two reanalyzes LONGTERM and ALL, the MATCH
simulation MFG and observed annual mean (see Supplement Sect. S2 and Figs. S2-S4 and
Table S1).
**3   Results**
**3.1   The performance of the model simulations and reanalyzes**
Before turning to the evaluation results, we investigate whether the two ozone reanalyzes
differ. We do this by comparing time series of annual $O_3$ metrics for the two data sets. The
investigation is presented in the Supplements and shows deviations in the latter years as the
number of sites in the ALL data set increases beyond the sites included in the LONGTERM
data set (see Supplement Sect. S3 and Figs. S2-S4). The deviation in annual mean near-
surface $O_3$ concentration is larger than for annual maximum 1 hour mean given that many of
the newer sites are sensitive to night-time inversions. Due to the visible deviation in results,
we use the LONGTERM for the trend and statistical analyzes in the paper, whereas both are
used for the evaluation of the 2dvar-method in this section. Both are included in the method
evaluation because the evaluation scores may be dependent on the density and specific
locations of the measurement sites. The ALL data set is to be used as a best estimate of
geographically resolved near-surface $O_3$ concentrations for Sweden for a subset period within
the full period 1990-2013.
In Table 2 we show the evaluation statistics from the validation of hourly near-surface $O_3$ in
2013. The near-surface $O_3$ concentrations from the MFG simulation compare well with





observations, and the 2dvar-technique leads to improvements. The spatially averaged
correlation coefficient of hourly near-surface $O_3$ concentrations (se Supplement Sect S1
increases from 0.67 when comparing the MFG $O_3$ concentrations to observations, to 0.76
when comparing the ALL reanalysis independently to observations through a cross validation
(Table 2). The %bias decreases from 1.4 to -0.3 and the RMSE is also improved in the
independent evaluation of the ALL reanalysis. Similar improvements are also obtained when
using fewer measurements (LONGTERM, Table 2), showing that the method is stable with
the number of measurement sites.   The cross validation spatial error (RMSE) is however
larger than that obtained when evaluating the MFG simulation against independent
observations, where the cross validation results indicates that the 2dvar reduces the quality of
the annual mean spatial variation in 2013. The lowest annual means in 2013 (Supplement
Table S2) are found in the sites Rödeby, Aspvreten, Östad, Norr Malma and Asa, where the
annual means are below 30 ppb(v). The highest annual means are found in Esrange, Norra
Kvill, Råö and Vavihill. This is likely caused by strong night-time inversions in the sites with
lower annual means. These night-time inversions depend to a large extent on local
topography, and are not uncommon in inland sites positioned at a low altitude in the local
landscape compared to the average of the surrounding area. This variation occurs at a higher
resolution than is captured by the MFG simulation (44km resolution). Simultaneously the
correction of the model by the data assimilation based on the differences between the model
and the measurements, results in readjustments of the model results for the surrounding area
and specifically for other sites not affected by night-time inversions. This is illustrated in the
Supplements (Fig. S5). Overall, the independent cross validation shows that the 2dvar method
improves the performance of the modelled hourly mean $O_3$ compared to the MFG simulation.
This is true not only in the measurement sites, but also elsewhere, with exception for the
spatial variation in annual mean.
In Fig. 5 we compare trends in annual near-surface $O_3$ percentiles over the period 1990-2013
for the MFG simulation, the LONGTERM reanalysis and the sum of contributions.
Investigating the additivity of the four contributions (bound, meteo, Se emis and Eur emis),
we compare the $O_3$ concentration trends in the MFG simulation to the trend in the sum of the
contributions (SUM, Fig. 5a). Almost all values fall close to the 1:1 line. Only a few of the
very weakest $O_3$ trends fall outside the factor 2 lines. Thus, the contribution experiment can
be used to explain the MFG $O_3$ trend. Comparing the LONGTERM and MFG trends in near-
surface $O_3$ (Fig. 5b), the values are within a factor of 2 for most percentiles and regions. There



is a general tendency for the positive MFG trends to be stronger than the reanalyzed trend
(LONGTERM). The largest deviations in the $O_3$ trends are in the North and the relationship
between these two is not as linear as in the other two regions. Most of these trends are
however not significant. This demonstrates the added value of the measurement model fusion,
where errors in the modelled trend are corrected by the analysis. The deviations are small
enough to conclude that in most cases the MFG is representative, showing that the MATCH
model can be used to understand the trends in the LONGTERM data set.
In conclusion we have shown that the MFG performs well for hourly near-surface $O_3$
concentration and the 2dvar analysis improves the performance to almost perfect
correspondence to the measurements in the measurement locations, and improved
performance elsewhere (cf. the cross-validation), with the exception of the spatial variation.
There is an added value of a reanalysis when investigating the trend of near-surface $O_3$
concentrations. The MATCH model can be used to investigate the causes to the reanalyzed $O_3$
trend. In the North the trends in the reanalyzed and the MFG $O_3$ concentration deviates by
more than a factor of 2 for some percentiles. We will focus on this deviation more in the final
discussion (Sect. 4).

## 3.2    Reanalyzed near-surface ozone in Sweden 1990-2013

The mean 1990-2013 seasonal variations in monthly mean and monthly maximum of 1h mean
near-surface $O_3$ are presented in Fig. 6, averaged over the three regions: North, Central and
South (as defined in Fig. 3). Spatially resolved statistics for annual mean and annual
maximum of 1h mean near-surface $O_3$ are provided in Fig. 7. Time series of annual
percentiles averaged over the three regions are shown in Fig. 8.

### 3.2.1    1990-2013 period statistics

The near-surface $O_3$ in Sweden exhibits a seasonal variation, which peaks during spring (Fig.
6). In the North the seasonal maximum concentration occurs in April, whereas it occurs later,
in May, in the regions further south. The earlier peak in the North, as compared to the South,
was also shown by Klingberg et al. (2009) for in situ observations. In the North, the seasonal
peak in monthly mean $O_3$ concentration is higher than the corresponding seasonal peaks in the
other two regions, and this is a feature throughout the whole winter half-year: the monthly





mean $O_3$ concentrations are higher in the North than the more southerly regions during Oct-
April. During the summer, the monthly means are higher in the South than in the other two
regions. This leads to a 24-year period mean value (Fig. 7) that is highest in the northerly
mountains and lowest in central Sweden. This pattern is also supported by Klingberg et al.
(2009) based purely on observations, but including a larger number of observation sites
through the inclusion of passive diffusion samplers.
For the period mean seasonal variation in monthly maximum 1h mean near-surface $O_3$ (Fig.
6b) there is a similar seasonal peak in April-May, but there is also a secondary peak during
summer (in August). The further south the higher is the monthly maximum 1h mean near-
surface $O_3$ during March-October. This applies to both the primary and the secondary
seasonal peaks in monthly maximum. The 24-year period mean of the annual maximum of 1h
mean near-surface $O_3$ (Fig. 7) is lower in central Sweden than in the South and the North, and
it is highest in the South.
The lower period mean of the near-surface $O_3$ in the South than in the North is possibly
caused by night-time inversions at some of the southerly sites, and also therefore the reason
for the opposite gradient for the annual maximum 1h mean as compared to the annual mean.
The difference in spatial pattern between the south, central and northern parts of Sweden is
why we choose the three regions defined in Fig. 3. The period maximum of the annual means
and period maximum 1h mean near-surface $O_3$ concentrations have similar spatial variation as
the period means (Fig. 7). The overall 24-year maximum 1h mean near-surface $O_3$ reaches
above 240 $\mu$g m$^{-3}$ in isolated parts of the South, and is generally above 180 $\mu$g m$^{-3}$ in the south
and 130 $\mu$g m$^{-3}$ in the central and northern part of Sweden.

### 3.2.2  Trend over the period

Seasonal variations are also present in the trend of both monthly mean and monthly maximum
1h mean near-surface $O_3$ concentrations (Fig. 6). Monthly means increase strongly during
winter and spring (approx. Nov-April), and decrease moderately (North) or strongly (Central
and South) during summer (May-Aug). The trends in monthly maximum 1h mean follow a
similar pattern. Generally, the rate of change is stronger or at the same level in the Central and
South as compared to the North. The strongest decrease is in the August maximum 1h mean
in the South and Central, and the strongest increase is in the March monthly mean. The day of
the year when the annual maximum 1h mean near-surface $O_3$ occurs shifts to earlier in the





year in the later part of the period, although there is large inter-annual variation, which is
stronger in the South than in the North (Supplements Fig. S6).
The annual mean near-surface $O_3$ (Fig. 7d,e) increases almost everywhere in Sweden over the
time period. The trend is however only significant in restricted parts of Central and South
regions, due to considerable inter-annual variation in the areas with the highest trend. The
annual maximum 1h mean near-surface $O_3$ (Fig. 7i,j) is significantly decreasing in South and
Central regions, whereas the change in the North is a mixture of increase and decrease, and it
is without significance in most areas. The decrease in the South and Central annual maximum
1h mean is a result of the strong decrease in the summer-time $O_3$ maximum; in the beginning
of the 24-year period, the southern summer-time maximum is more often the annual peak
rather than the spring-time maximum, whereas the summer-time maximum is more often
secondary to the annual maximum in the end of the period. The annual maximum 1h mean is
shifted to earlier in the year (Supplements Fig. S6). In a study of four rural European sites and
one in western United States, Parrish et al. (2013) showed that not only are springtime $O_3$
concentrations larger in recent years than in earlier decades, but also that the seasonal
maximum now also occurs earlier, as in our results for Sweden. This change in seasonal cycle
is also supported by the work by Cooper et al. (2014). The change in the annual maximum 1h
mean near-surface $O_3$ from summer-time peak to spring-time peak means that more than one
process can be the cause of the change (increasing spring-time and decreasing summer-time).
We proceed by investigating the trend in annual percentiles of hourly near-surface $O_3$
concentration, averaged[2] over the three Swedish regions (cf. Fig. 3). The temporal evolution
of 11 percentile levels from the 0th (annual minimum 1h mean) to the 100th (annual maximum
1h mean) are shown in Fig. 8, and the corresponding trends with indication of significance
levels are recaptured in the Supplements (Table S3). In all three regions the low and medium
percentiles are increasing, while the highest percentiles are decreasing from 1990 until 2013.
This was also shown by Simpson et al. (2014) based on observations for northern Europe and
based on observations for Europe, US and East Asia by Lefohn et al. (2017). Further, using
hourly $O_3$ observations, Karlsson et al. (2017) showed that reduced concentrations were

---

[2] The percentile is calculated per grid square for all hours in each year, then regional mean annual percentiles are calculated and finally the trend is calculated based on these averaged percentiles.





restricted to the highest $O_3$ concentrations during summer daytime, while the increase in low
and mid-range concentrations occurred during wintertime at both day and night.
In Central and South regions the decrease in the highest near-surface $O_3$ percentiles are
stronger than in the North, and significant and this decrease is evident throughout the
maximum 10% percentile range (although the change is not significant for the 90[th] and 95[th]
percentile levels; cf. Fig 3). This change is mainly caused by decreased high values during the
summer-time. In the North, only the annual maximum 1h mean is decreasing and the inter-
annual variability is stronger than the rate of change, indicated by the lack of significance for
this percentile. The medium and low percentile increase in the North is moderate, but
significant, for most percentiles up to the 95[th], with very similar rates of change. In the
Central and South the change in the low percentiles is highly significant and stronger than in
the North. This is an indication that the increase in low near-surface $O_3$ concentrations cannot
only be explained by increasing background. As a result of the decrease of high and increases
of low percentiles, there has been a narrowing of the range of the near-surface $O_3$
concentrations over the period. This was also observed in the US by Simon et al. (2015) for
1998-2013, studying urban and regional background measurements across the US. They
interpret this as a response to the substantial decrease in $O_3$ precursor emissions in the US
over the time period. Decreased primary NO emissions results in decreased $O_3$ titration close
to combustion sources, but also reduces local $O_3$ further away from the emissions sources
under weather states favorable for $O_3$ formation. In the next section we investigate the impact
of Swedish and European emission decrease over the period, and relate this to the impact of
change in the chemical composition of the hemispheric background and meteorological
variations.
**3.3   Attribution of the change in near-surface ozone**
In this section we quantify the contributions of physical processes to the modelled trend of
near-surface $O_3$ concentration in Sweden during the period 1990-2013. We investigate the
impact of the trend in lateral and upper boundaries, meteorological variations and Swedish
and European (non-Swedish, full domain) anthropogenic emission change. In Figs. 9 and 10
the contributions to the trend in seasonal variations and percentiles are quantified for the
North and South regions.



We start our attribution by analyzing the impact of changing hemispheric background levels
of relevant chemical species ("bound" bars in Figs. 9 and 10). These show an increase in
monthly mean and maximum 1h mean throughout the year and for all percentiles, mainly as a
result of our assumption of an increasing $O_3$ concentration trend in the lateral and upper
boundaries during the 1990s and constant boundary conditions for $O_3$ during the rest of the
period. There is a seasonal variation in the trend of the boundary contribution, with a
minimum during summer. This variation is likely a result of an $O_3$ destruction process that is
stronger during summer than winter, such as dry deposition to vegetation and photolysis of
ozone. The seasonal variation in the contribution to the trend from the boundary impacts both
monthly mean and maximum 1h mean. Our representation of the trend in the concentration of
species at the model domain boundary is climatological. The climatological upper boundary
means that the inter-annual variations in near-surface $O_3$ are likely underestimated in remote
locations. The impact on inter-annual variations may be largest at high altitudes or far away
from the major anthropogenic sources. Hess and Zbinden (2013) showed the importance of
the stratospheric contribution to the inter-annual variation at Mace Head and Jungfraujoch; it
is possibly also important in the north of Sweden, especially in the mountainous areas. Such
variation is not captured by the boundary settings, but it is indirectly included in the
reanalyzes data sets through the variation in the measurements included in the data
assimilation. As a consequence, the MFG and "bound"  simulations underestimate the inter-
annual variability as compared to observations and the reanalysis (cf. Table 2), and this could
also affect the "bound" trend.
The impact of meteorological low-frequency variations ("meteo") during the 24 years is also
an important factor, but more difficult to interpret. The meteorological variation acts to cause
a positive trend in near-surface $O_3$ concentration for most monthly means and maxima, as
well as for most percentiles. The meteorological influence on the trend is as large as the
impact of the change in boundary, for most percentile levels in the South, while it is weaker
for most percentile levels in the North.
During the period 1990-2013 both European (full domain, non-Swedish) and Swedish
emissions have decreased strongly. There is a strong seasonality in the impact of the
decreasing European emissions, and the contribution to the trend of the Swedish emissions
follows the same pattern but with smaller magnitude (cf. Fig. 9, "Eur emis" and "Se emis"
respectively). During summer the decreasing emissions have acted to lower both the monthly





mean and maximum 1h mean. During winter the trend in monthly maximum 1h mean is
unaffected by the change in emissions, indicating that the highest near-surface $O_3$
concentrations during winter are due to other sources than local $O_3$ production. Emission
decreases have acted to cause increases in monthly mean near-surface $O_3$ concentrations in
the winter, due to reduced $O_3$ destruction by primary NO emission. Trends in percentiles (Fig.
10), show that the emission decrease has caused decreases to percentiles higher than the $50^{th}$
level, and increases below. The impact is stronger in the South than in the North, which is
expected due to the South being closer to the European continent. The contribution of the
trend in emissions is often stronger than the changing boundary, e.g. in the South for most
percentiles and for monthly maximum 1h mean during the summer half-year in both regions.
Thus, the observed increase in low and medium near-surface $O_3$ levels is caused by a mixture
of both changes to the hemispheric background levels and emission reductions of $O_3$
precursors, while the decrease in the high percentile levels is mainly caused by emission
decrease.
**3.4   Implications for health and vegetation impacts**
For the protection of vegetation, the target value by EU (EU directive 2008/50/EC) states that
the 5-year mean AOT40 (near-surface $O_3$ concentration above 40 ppb(v) accumulated over
May-July; AOT40c) must not exceed 9 ppm(v) h, and as a long-term goal AOT40c must not
exceed 3 ppm(v) h during a calendar year.  For protection of human health the target value by
EU (EU directive 2008/50/EC) states that the daily maximum running 8 hour mean near-
surface $O_3$ concentration must not exceed 120 $\mu$g m$^{-3}$ more than 25 days per year as a 3-year
mean, and as a long-term goal the daily maximum of 8h mean near-surface $O_3$ concentration
must not exceed 120 $\mu$g m$^{-3}$ at all. Sweden has formulated 16 environmental quality
objectives, including clean air, alongside specifications to help reach these objectives. The
following specifications are currently valid for near-surface $O_3$ concentration in Sweden (NV,
2015): the hourly mean must not exceed 80 $\mu$g m$^{-3}$, the daily maximum 8h mean must not
exceed 70 $\mu$g m$^{-3}$ and AOT40f ($O_3$ concentration above 40 ppb(v) accumulated over April-
September) must not exceed 5 ppm(v) h. In Table 3 we present the linear trends in our
reanalysis data set for these metrics, and have collected geographically resolved statistics,
such as the period mean, maximum and linear trend in the Supplements (Figs. S7-S11).





The narrowing of the $O_3$ concentration range, especially through increasing lower percentiles,
can impact human and vegetation exposure to $O_3$. The effect metrics based on accumulation
of values above a threshold (AOT40c; AOT40f; SOMO35, Sum of Ozone Means[3] Over 35
ppb(v)) and the number of days with daily maximum of 8h mean near-surface $O_3$
concentration exceeding 120 $\mu g\ m^{-3}$ have been decreasing over the period in the South and
Central regions, as have the highest values in the year. This is in accordance with the decrease
in the highest percentiles in these regions (cf. Supplements Table S3). Conversely, the metrics
with lower threshold values increase, such as the number of hours exceeding 80 $\mu g\ m^{-3}$ and
the number of days with daily maximum 8h mean near-surface $O_3$ concentration exceeding 70
$\mu g\ m^{-3}$. This increase is significant in the North, whilst it is not significant in the South and
Central. This agrees with the change in medium and low percentiles. A continued increase in
low values would cause a continued increase in these metrics, and would eventually reverse
the decreasing trend to an increase. This is valid specifically for those metrics with
accumulation of values or higher thresholds, such as SOMO35 and AOT40c.
The highest near-surface $O_3$ concentrations, associated with short-term (acute) health impacts,
show a clear and significant decrease in the South (where the highest values occur), leading to
an improvement in health impacts. For long-term health effects, there is no established
threshold below which there are no adverse effects, even if SOMO35 often is used. The
increase in low values (and e.g. the annual mean) has negative impacts on health, although
SOMO35 is decreasing in the South and Central Sweden. This increase is also of concern
given that policy choices will cause further reductions in local NO emissions – which are
highly correlated to where people reside – thus increasing the sensitivity of $O_3$ to the
background and hemispheric background level. Despite this, the solution is not to reverse
policies that reduce local NO production, given that this would negatively impact both the
highest values and the hemispheric background. A solution must therefore be sought via
international policy regulations.

---

[3] For SOMO35 the Mean is defined as the daily maximum of running 8h mean near-surface $O_3$ concentrations
and the accumulation is over a year unless otherwise is stated.




## 4 Discussion

This work improves upon previous studies by investigating the trends in near-surface $O_3$ concentration via a combination of both observed and modelled knowledge. The respective advantages of modeling (geographical and temporal coverage) and observations (the most reliable $O_3$ concentration estimate at a discreet point) can be exploited through data assimilation to reach a greater understanding of the atmospheric state, and the model can further be used as a tool to explain what is described.

Our results should, however, also be viewed in the context of their limitations. The model simulations have a relatively coarse horizontal resolution, meaning that processes that are more local in origin are not captured by the model – these include the role of local topography or coastal climate for the night-time boundary layer stability (Klingberg et al., 2011), or local emission sources. As a result, the data assimilation scheme will spread such features to parts of the model where they are not valid. Some of the southerly sites in the data assimilation are known to experience night-time inversions and the reanalysis will thus be affected by this. We choose however not to exclude these data from the data assimilation, on the basis that this is restricted to occasional events during night-time. An improvement in the spatial resolution of the model would improve the spatial representation of the analysis, since the difference between observation and model has the potential to decrease at these observation sites.

As with all modeling studies, the model cannot perform better than the quality of the forcing input data. Knowledge of emissions in the beginning of the 24-year period is less comprehensive than at the end, which could introduce artificial trends to the MFG. The trends in lateral and boundary conditions are taken from the work by Engardt et al. (2017) and are based on observed trends at the lateral boundaries of Europe. The upper boundaries are especially poorly represented, and as a consequence so is the stratospheric contribution to the inter-annual variation and trend. The data assimilation reduces the impact of errors in the lateral and upper model boundaries. However, the reanalysis may still be affected in regions with sparse measurement coverage. This can affect the attribution to the trend. In this study the MFG simulation captures the observed (reanalyzed) trend reasonably well, but there is a discrepancy between the reanalysis and MFG trend for most percentile levels in North Sweden. To investigate this in more detail, we have compared the error in trend by percentile (the difference between the trends in MFG and LONGTERM) to the trend caused by the four contributions (bound, meteo, Se emis and Eur emis). The resulting figure is included in the





Supplements (Fig. S12). There is a 1:1 relation between the impact of the trend in the European emissions and the deviation between the MFG and the LONGTERM trends. This could be caused by overestimation of the European emissions trend. A similar tendency is seen for the Swedish emission contribution in the Central and South regions. This calls for emission inventories to be improved in order to make sure the trend in ozone precursor emissions is correct. Another reason for this could be too strong model sensitivity to the European emission trend in the North. If this was true, it would have implications for sensitivity studies that consider the future development of near-surface $O_3$. In studies relating the impacts of future climate change to future anthropogenic precursor emission change, a robust conclusion for most models is that the impact on near-surface $O_3$ concentration of future precursor emissions is much stronger than the impact of climate change (e.g. Engardt et al., 2009, Langner et al, 2012; Watson et al., 2016). If models are too sensitive to trends in emissions in remote areas, compared to other processes, such a conclusion might change. Parrish et al. (2014) also compared observed and modelled trends and found that the three chemistry-climate models studied failed to reproduce the observed trends – the modelled $O_3$ concentration trend was approximately parallel to the estimated trend in anthropogenic precursor emissions of $NO_x$, whilst observed $O_3$ concentration changes increased more rapidly than these emission estimates. This implies that there is a lack of knowledge relating to controls of concentrations of tropospheric $O_3$. Whether it is the trend in ozone precursor emissions or the model sensitivity to emissions that need improving is left for future studies.

Finally, we conducted a trend analysis of the reanalyzed near-surface $O_3$ using linear regression. We have chosen to present the trend in the LONGTERM data set in all analyzes, regardless of whether it is statistically significant or not. We stress that a trend contains valid information even where it is not statistically significant – and it will become significant if the change and variability remains the same over time. We also recognize that there are other methods of investigating the statistical behavior of the data set, and therefore welcome further use of the data, which may be accessed upon request from the corresponding author.

## 5    Conclusions

- We have constructed two hourly reanalyzes of near-surface $O_3$ for Sweden for the period 1990-2013: one time-consistent reanalysis and one using all available hourly





measurements. Both data sets are available upon request from the corresponding
author.
• We have evaluated the performance of the reanalyzed near-surface $O_3$ and mainly
found improved performance compared to the MATCH model.
• Our results show:
○ High near-surface $O_3$ concentrations in Sweden are decreasing and low $O_3$
concentrations are increasing.
○ Health and vegetation impacts due to high near-surface $O_3$ concentrations
(quantified by policy related threshold metrics) have decreased in Sweden as a
result of the decrease in the highest ozone values.
○ Decreasing emissions in Europe have led to decreasing summer-time near-
surface $O_3$ concentrations, as well as a decrease of the highest concentrations.
○ The rising low concentrations of near-surface $O_3$ in Sweden are caused by a
combination of rising hemispheric background $O_3$ concentrations,
meteorological variations and $O_3$ response to European $O_3$ precursor emission
regulation.
○ There is a discrepancy between modelled and observed (reanalyzed) $O_3$ trends
in northern Sweden. This could be caused by erroneous trends in the historical
anthropogenic ozone precursor emissions used here or that our model is too
sensitive to changes in emissions. If the latter is true, it implies that the
evolution of future precursor emissions may have a smaller impact on future
near-surface $O_3$ concentrations than shown by earlier studies.
**Acknowledgements**
This project was funded by the Swedish Environmental Protection Agency (EPA), through
funding directly to the reanalysis (contract no. 2251-14-016) and through the research
program Swedish Clean Air and Climate (SCAC) and NordForsk through the research
programme Nordic WelfAir (grant no. 75007). The annual mapping with the MATCH
Sweden system is funded by the Swedish EPA.





1     Thank you to Sverre Solberg (NILU, Norway) for all help, especially with the selection of

2     Norwegian observation sites.




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



**Figure and table legends**
**Figure legends**
Figure 1. A flow-chart of the relevant part of the MATCH Sweden system for this reanalysis
study.
Figure 2. (a) Secular trend of factors used for scaling boundary concentration of relevant
species. Note that the hemispheric background ozone concentrations are assumed constant
from 2000 onwards. CO and NMVOC boundaries are held constant throughout the
simulation. (b) Temporal trend of total domain (solid lines; left vertical scale) and Swedish
(dashed lines; right vertical scale) annual $O_3$ precursor emissions utilized by MATCH from
1990 to 2013. Emissions of nitrogen oxides (NOx), non-methane volatile organic compounds
(NMVOC), carbon monoxide (CO) and biogenic isoprene (C5H8) are indicated by different
colors (cf. legend); emissions of sulfur oxides ($SO_x$) and ammonia ($NH_3$) are excluded from
the panel.
Figure 3. Instrumentation sites for hourly near-surface ozone concentration observations in
Sweden and Norway, which are used in the variational analysis. Red circles: sites with full
data coverage. Blue circles: sites with restricted data coverage. The subdivision of Sweden
into three regions (North, Central and South) follows county borders, as indicated by the fat
black lines.
Figure 4. Data availability at instrumentation sites for hourly near-surface ozone
concentration observations in Sweden and Norway. Red squares: years with at least 80 %
annual data for sites with full data coverage (see also Fig. 3). Light red: sites with <80 %
annual data (data capture indicated in square) for sites with full coverage. Blue and light blue
squares: as for the red squares, but for sites with restricted data coverage.
Figure 5. Trends in near-surface ozone percentile levels averaged for the three regions North
(blue), Central (green) and South (magenta) for the sum of the contributions to the trend
(SUM) vs the MATCH model simulation MFG (a) and the MATCH model simulation MFG
vs the reanalysis LONGTERM (b). Filled circles indicate significant trends ($p \leq 0.05$) in the
MFG simulation, whereas non-significant MFG trends ($p > 0.5$) are indicated by an empty
circle. 1:1 line in black, factor 2 lines in dark grey and equal sign quadrants are separated by
light grey lines.



Figure 6.  Seasonal cycle of monthly mean (a) and monthly maximum of 1h mean (b) near-
surface ozone concentrations averaged over the period 1990-2013 (solid lines; left vertical
scale) and the linear trend over the same period of the respective monthly values (dashed
lines; right vertical scale) in the three regions North, Central and South Sweden (cf. Fig. 3).
The corresponding regions are referred to by different colors, see legend. Results from the
LONGTERM reanalysis.
Figure 7. Statistical properties of the annual mean (top row; (a)-(e)) and annual maximum 1h
mean (bottom row; ((f)-(j)) near-surface ozone concentration. In the columns from left to
right: 1990-2013 mean ((a),(f)), 1990-2013 maximum ((b),(g)), 1990-2013 standard deviation
((c),(h)), linear trend over the period 1990-2013 ((d),(i)) and significance in the linear trend
over the period ((e),(j)). Results from the LONGTERM reanalysis.
Figure 8. Temporal variation of annual percentiles of near-surface ozone concentrations
averaged over the three regions North (a), Central (b) and South (c) of Sweden (cf. Fig. 3).
The line marked 0 is the zero-percentiles (lowest hourly mean near-surface ozone
concentration of the year), 100 is 100-percentile (highest hourly mean near-surface ozone
concentration of the year), 50 is the 50-percentile (i.e. annual median of the hourly mean near-
surface ozone concentration). The sign of the corresponding linear trend (cf. Supplements
Table S3, including a statistical analysis of the trend) of each percentile is indicated by colour:
a negative linear trend over 1990-2013 is indicated by grey symbols; a positive trend by
orange symbols. Statistically significant trends ($p \leq 0.05$) are indicated by thick lines. Results
from the LONGTERM reanalysis.
Figure 9. Linear trend over 1990-2013 in monthly mean ((a),(c)) and monthly maximum 1
hour mean ((b),(d)) near-surface ozone concentration for the North ((a),(b)) and the South
((c),(d)) Swedish regions (cf. Fig. 3). Reanalyzed (white diamond; LONGTERM reanalysis)
and modelled "first guess" (MFG) near-surface ozone trend (green diamond), and modelled
contributions to the near-surface ozone trend due to change in emissions:  anthropogenic
Swedish (dark blue, Se emis) and full domain, non-Swedish (fair blue, Eur emis), emissions,
trend in top and lateral boundaries of relevant species (yellow, bound) and variation in
meteorology (brown, meteo). The sum of the modelled contributions is indicated by the
dashed green line.
Figure 10. Linear trends over 1990-2013 in annual percentiles of hourly mean near-surface
ozone concentrations for the North (a) and the South (b) Sweden regions. Reanalyzed (white



diamond; LONGTERM reanalysis) and modelled MFG near-surface ozone trend (green
diamond), and modelled contributions to the near-surface ozone concentration trend due to
change in emissions: anthropogenic Swedish  (dark blue, Se emis) and full domain, non-
Swedish (fair blue, Eur emis), emissions, trend in top and lateral boundaries of relevant
species (yellow, bound) and variation in meteorology (brown, meteo). The sum of the
modelled contributions is indicated by the dashed green line.



**Table legends**
Table 1a. Model calculations and scenarios, all covering the years 1990-2013, including the
"first guess" to the data assimilation and base case to the sensitivity simulations (MFG), two
reanalysis data sets (LONGTERM and ALL), sensitivity scenarios (MEUR, MSE, MBC and
MMET).
Table 1b. Formation of contributions to the linear trend over the period 1990-2013 from the
sensitivity simulations (Se emis, Eur emis, Bound and Meteo, see Table 1a).
Table 2. Evaluation of modelled hourly near-surface ozone concentrations in 2013 at Swedish
observation sites. Mean value (mean), standard deviation ($\sigma$), model mean bias normalized by
the observed mean (%bias), Pearson correlation coefficients (r) for data including at least 10
pairs, the root mean square error (RMSE) and number of observed hours at the sites. The
evaluation includes the reanalyzed data sets ALL and LONGTERM, where ALL is evaluated
at the 12 Swedish sites included in that simulation, and LONGTERM is evaluated at the 6
Swedish sites included in that simulation (cf. Fig. 4). For each of these data set evaluations we
include the observation *dependent* reanalysis (2dvar), the observation *independent* cross
validation of the reanalysis (cross) and the MATCH base case simulation (MFG). The top half
of the table shows the temporal performance (spatial mean of statistics, see Supplement Sect.
S1). The bottom half of the table shows spatial performance (spatial statistics of annual
means, see Supplement Sect. S1).
Table 3. Linear trend during 1990-2013 of policy related metrics in the 3 Swedish regions
North, Central and South (cf. Fig. 3). Stars (*, **, and ***) indicate that the trend is
significant ($p \leq 0.05$, $p \leq 0.01$, $p \leq 0.001$, respectively).



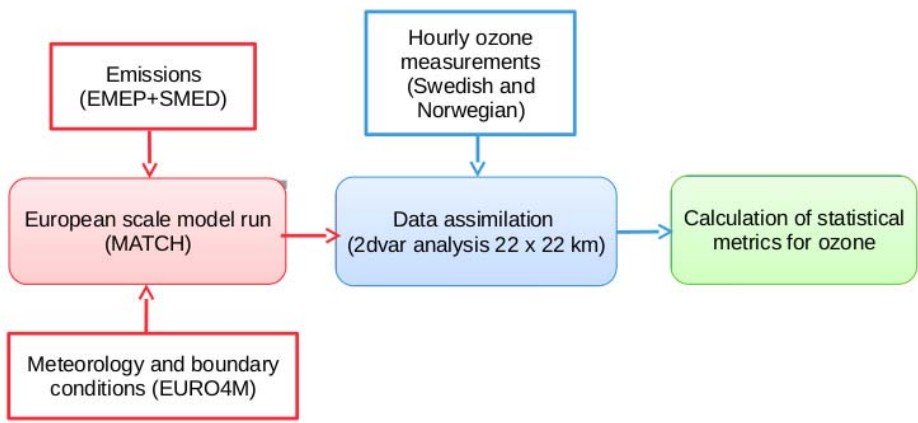

2    Figure 1. A flow-chart of the relevant part of the MATCH Sweden system for this reanalysis

3    study.

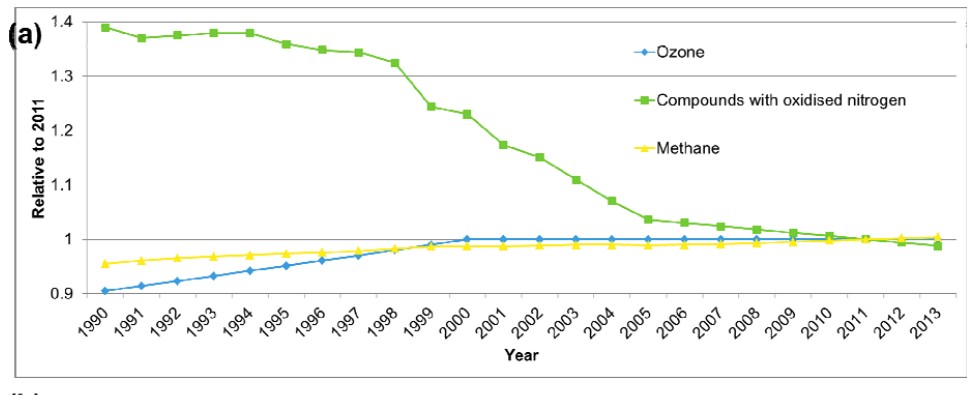

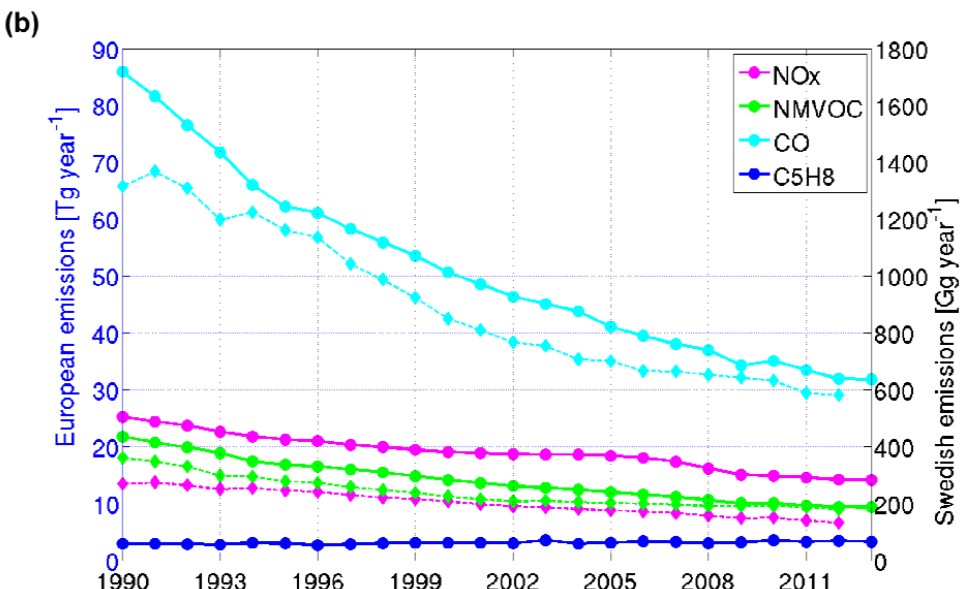

Figure 2. (a) Secular trend of factors used for scaling boundary concentration of relevant species. Note that the hemispheric background ozone concentrations are assumed constant from 2000 onwards. CO and NMVOC boundaries are held constant throughout the simulation. (b) Temporal trend of total domain (solid lines; left vertical scale) and Swedish (dashed lines; right vertical scale) annual $O_3$ precursor emissions utilized by MATCH from 1990 to 2013. Emissions of nitrogen oxides (NOx), non-methane volatile organic compounds (NMVOC), carbon monoxide (CO) and biogenic isoprene (C5H8) are indicated by different colors (cf. legend); emissions of sulfur oxides ($SO_x$) and ammonia ($NH_3$) are excluded from the panel.



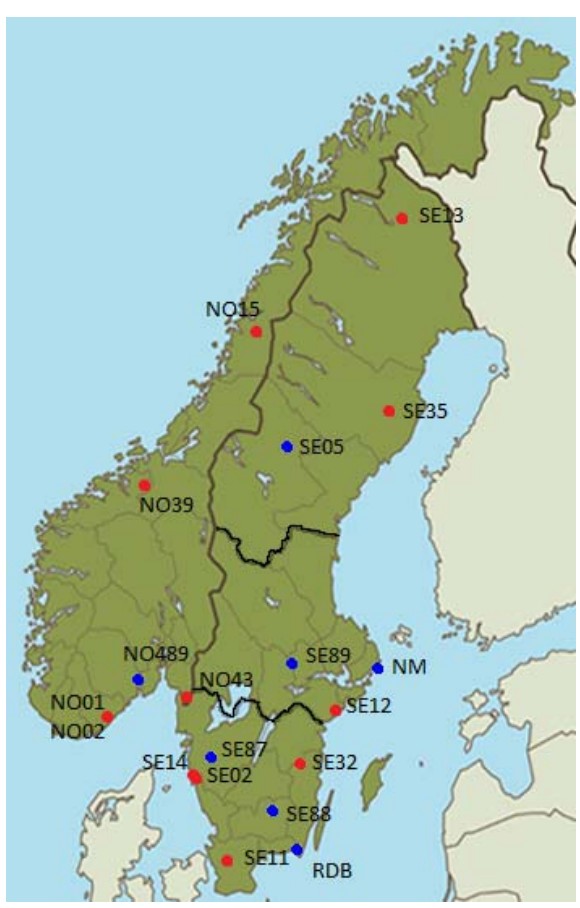

Figure 3. Instrumentation sites for hourly near-surface ozone concentration observations in
Sweden and Norway, which are used in the variational analysis. Red circles: sites with full
data coverage. Blue circles: sites with restricted data coverage. The subdivision of Sweden
into three regions (North, Central and South) follows county borders, as indicated by the fat
black lines.



| | | 1990 | 1991 | 1992 | 1993 | 1994 | 1995 | 1996 | 1997 | 1998 | 1999 | 2000 | 2001 | 2002 | 2003 | 2004 | 2005 | 2006 | 2007 | 2008 | 2009 | 2010 | 2011 | 2012 | 2013 |
|---|---|---|---|---|---|---|---|---|---|---|---|---|---|---|---|---|---|---|---|---|---|---|---|---|---|
| SE13 | Esrange | 30 | | | | | | | | | | | | | | | | | | | | | | | |
| SE35 | Vindeln | | | | | | | | | | | | | | | | | | | | | | | | |
| SE05 | Bredkälen | | | | | | | | | | | | | | | 58 | | | | | | | | | |
| SE89 | Grimsö | | | | | | | | | | | 42 | | | | | | | | | | | | | |
| NM | Norr Malma | | | | | | | | | | | | | | | 0.01 | | | | | | | | | |
| SE12 | Aspvreten | | | | | | | | | | | | | | | | | | | | | | | | |
| SE32 | Norra Kvill | | | | | | | | | | | | | | | | | | | | | | | | |
| SE88 | Asa försökspark | | | | | | 50 | | | | 75 | | | | | 45 | | | | | | | | 58 | |
| SE87 | Östad | | | | | | | | | | | | 47 | 34 | 20 | 45 | 49 | 50 | 50 | 50 | 50 | 50 | 49 | 50 | |
| SE02 | Rörvik | | | | | | | | | | | | | | | | | | | | | | | | |
| SE14 | Råö | | | | | | | | | | | | | | | | | | | | | | | | |
| RDB | Rödeby | | | | | | | | | | | | | | | | | | | | | | | | 56 |
| SE11 | Vavihill | | | | | | | | | | | | | | | | | | | | | | | | |
| NO15 | Tustervatn | | | | 73 | | | | | | | | | | | | | | | | | | | | |
| NO39 | Kårvatn | | | | | | | | | | | | | | | | | | | | | | | | |
| NO489 | Haukenes | | 47 | 22 | 42 | 51 | 51 | 53 | 55 | 49 | 49 | 51 | 39 | 53 | 53 | | | | 40 | | 67 | | 72 | | |
| NO43 | Prestebakke | | | 65 | | | | | | | | | | | | | | | | | | | | | |
| NO01 | Birkenes I | | | | | | | | | | | | | | | | | | | | | | | | |
| NO02 | Birkenes II | | | | | | | | | | | | | | | | | | | | | | | | 79 |

Figure 4. Data availability at instrumentation sites for hourly near-surface ozone concentration observations in Sweden and Norway. Red squares: years with at least 80 % annual data for sites with full data coverage (see also Fig. 3). Light red: sites with <80 % annual data (data capture indicated in square) for sites with full coverage. Blue and light blue squares: as for the red squares, but for sites with restricted data coverage. White squares: no observations are available for that year and site. The LONGTERM reanalysis includes the red measurement sites, the ALL reanalysis includes both red and blue.



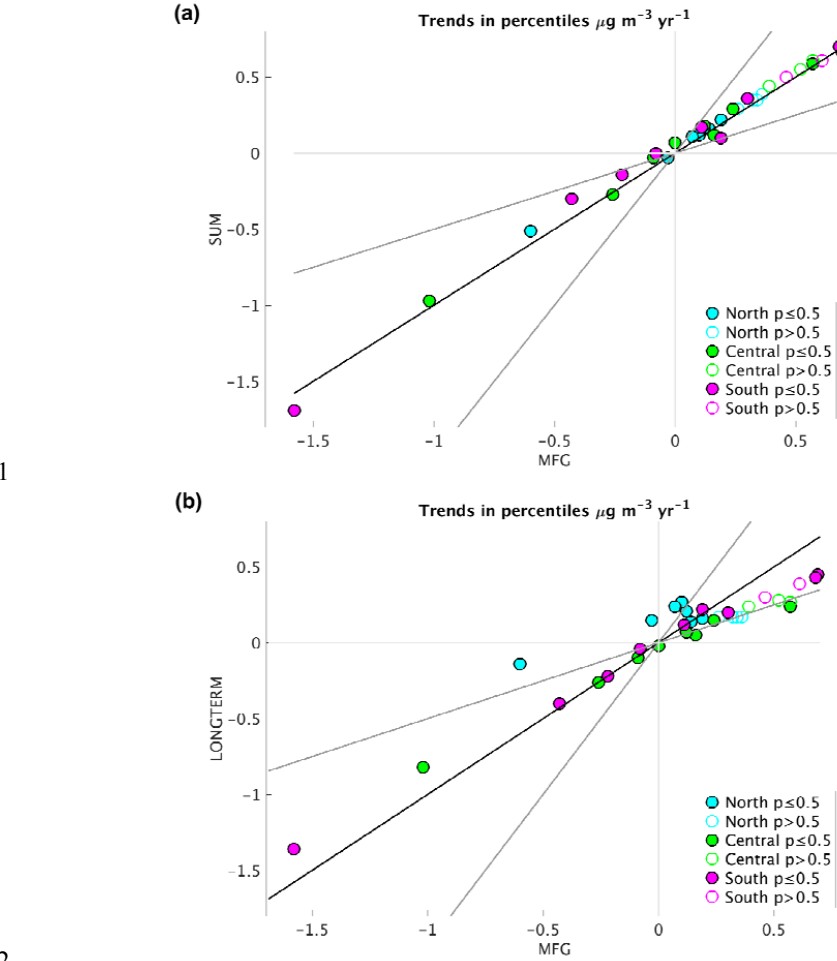

Figure 5. Trends in near-surface ozone percentile levels averaged for the three regions North
(blue), Central (green) and South (magenta) for the sum of the contributions to the trend
(SUM) vs the MATCH model simulation MFG (a) and the MATCH model simulation MFG
vs the reanalysis LONGTERM (b). Filled circles indicate significant trends ($p \leq 0.05$) in the
MFG simulation, whereas non-significant MFG trends ($p > 0.5$) are indicated by an empty
circle. 1:1 line in black, factor 2 lines in dark grey and equal sign quadrants are separated by
light grey lines.





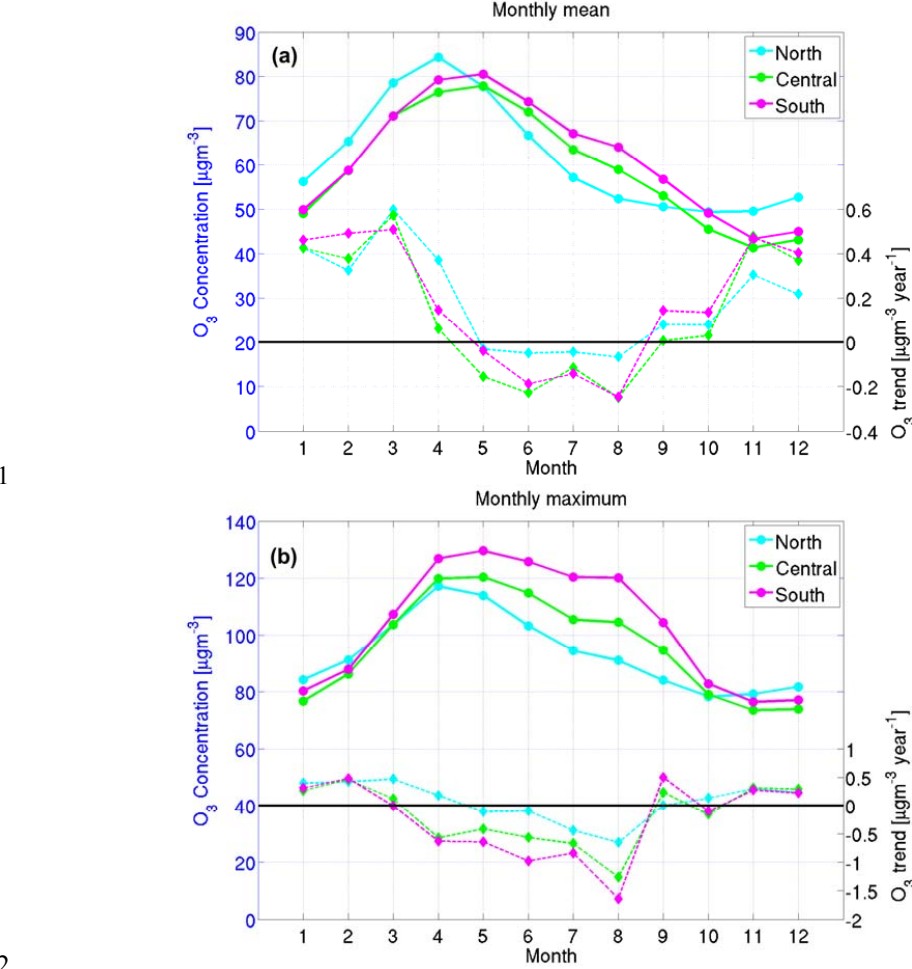

3 Figure 6. Seasonal cycle of monthly mean (a) and monthly maximum of 1h mean (b) near-

4 surface ozone concentrations averaged over the period 1990-2013 (solid lines; left vertical

5 scale) and the linear trend over the same period of the respective monthly values (dashed

6 lines; right vertical scale) in the three regions North, Central and South Sweden (cf. Fig. 3).

7 The corresponding regions are referred to by different colors, see legend. Results from the

8 LONGTERM reanalysis.





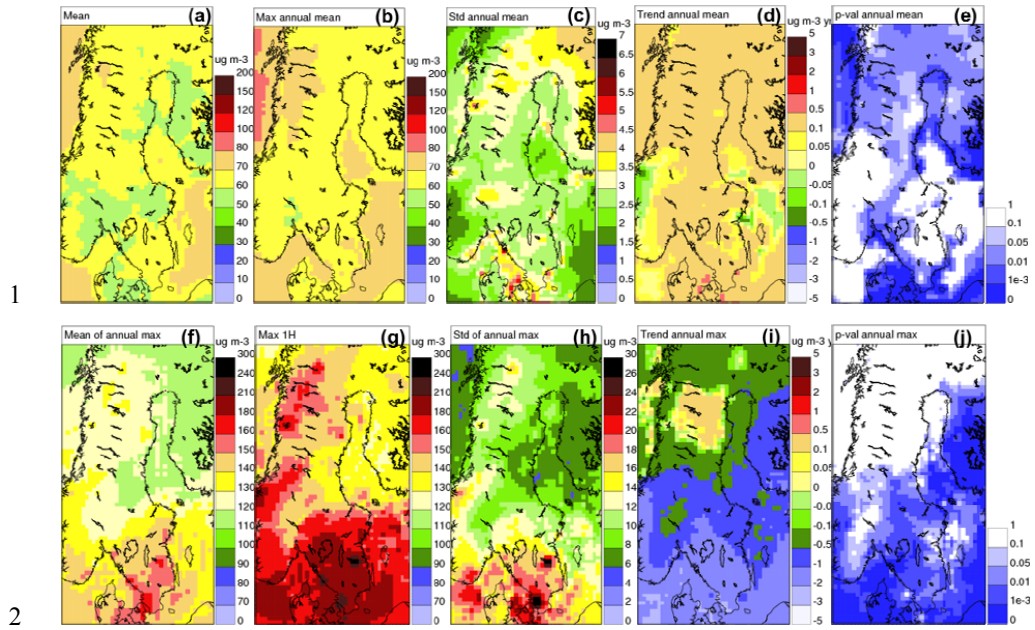

Figure 7. Statistical properties of the annual mean (top row; (a)-(e)) and annual maximum 1h
mean (bottom row; ((f)-(j)) near-surface ozone concentration. In the columns from left to
right: 1990-2013 mean ((a),(f)), 1990-2013 maximum ((b),(g)), 1990-2013 standard deviation
((c),(h)), linear trend over the period 1990-2013 ((d),(i)) and significance in the linear trend
over the period ((e),(j)). Results from the LONGTERM reanalysis.

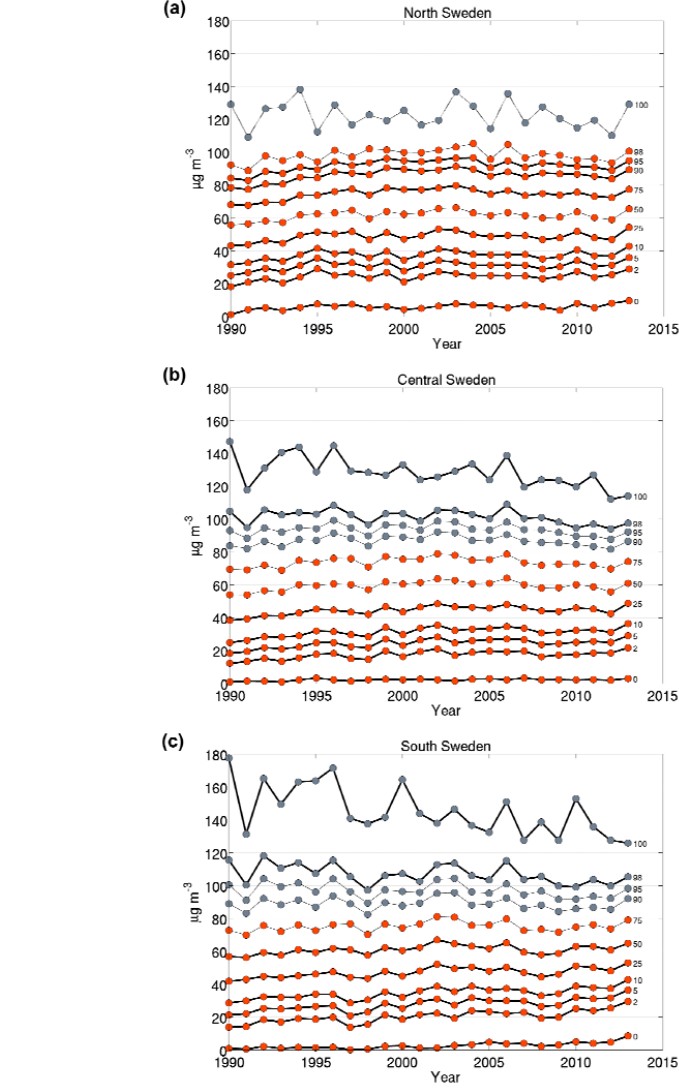

Figure 8. Temporal variation of annual percentiles of near-surface ozone concentrations
averaged over the three regions North (a), Central (b) and South (c) of Sweden (cf. Fig. 3).
The line marked 0 is the zero-percentiles (lowest hourly mean near-surface ozone
concentration of the year), 100 is 100-percentile (highest hourly mean near-surface ozone
concentration of the year), 50 is the 50-percentile (i.e. annual median of the hourly mean near-
surface ozone concentration). The sign of the corresponding linear trend (cf. Supplements
Table S3, including a statistical analysis of the trend) of each percentile is indicated by colour:





1    a negative linear trend over 1990-2013 is indicated by grey symbols; a positive trend by

2    orange symbols. Statistically significant trends (p≤0.05) are indicated by thick lines. Results

3    from the LONGTERM reanalysis.




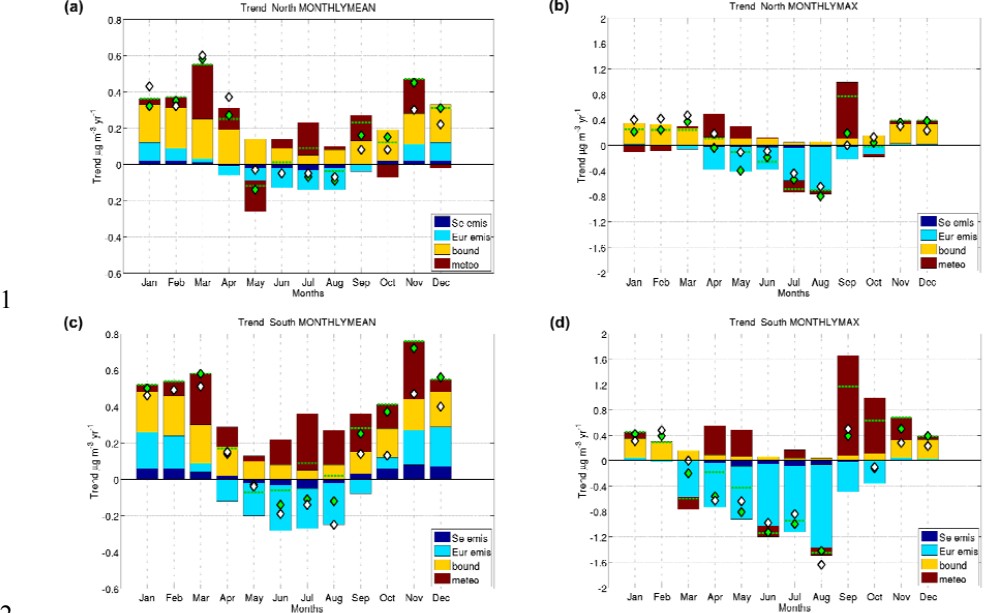

Figure 9. Linear trend over 1990-2013 in monthly mean ((a),(c)) and monthly maximum 1
hour mean ((b),(d)) near-surface ozone concentration for the North ((a),(b)) and the South
((c),(d)) Swedish regions (cf. Fig. 3). Reanalyzed (white diamond; LONGTERM reanalysis)
and modelled "first guess" (MFG) near-surface ozone trend (green diamond), and modelled
contributions to the near-surface ozone trend due to change in emissions:   anthropogenic
Swedish (dark blue, Se emis) and full domain, non-Swedish (fair blue, Eur emis), emissions,
trend in top and lateral boundaries of relevant species (yellow, bound) and variation in
meteorology (brown, meteo). The sum of the modelled contributions is indicated by the
dashed green line.



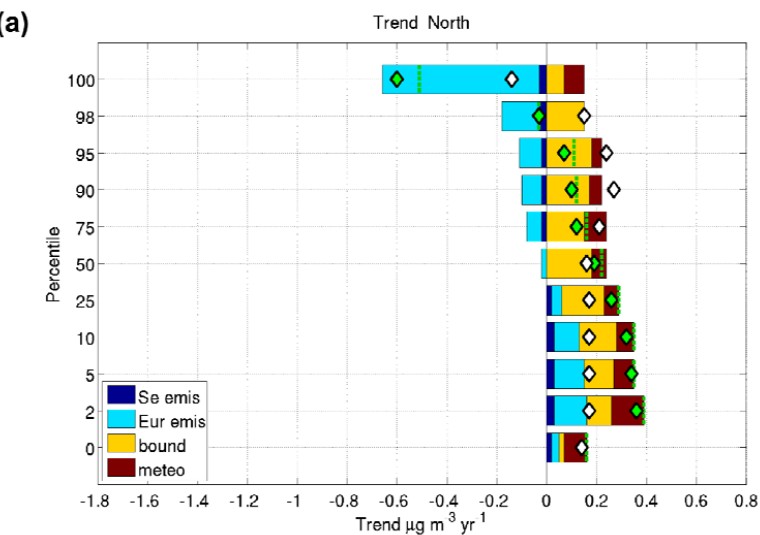

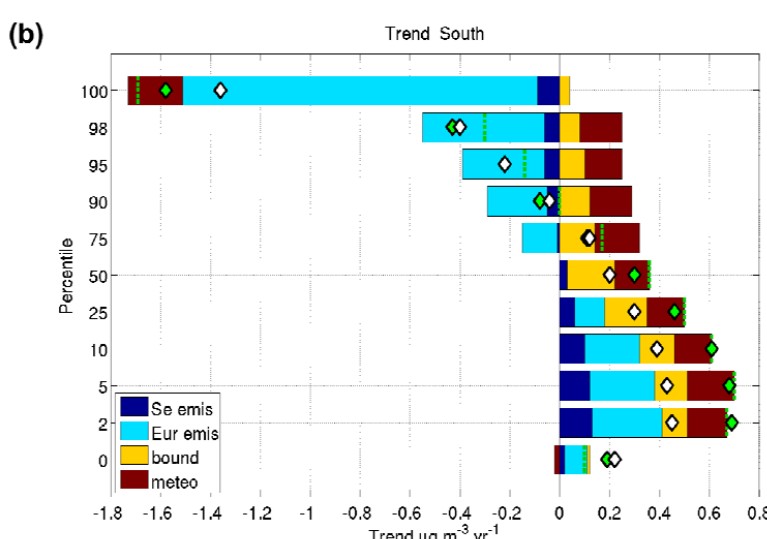

Figure 10. Linear trends over 1990-2013 in annual percentiles of hourly mean near-surface
ozone concentrations for the North (a) and the South (b) Sweden regions. Reanalyzed (white
diamond; LONGTERM reanalysis) and modelled MFG near-surface ozone trend (green
diamond), and modelled contributions to the near-surface ozone concentration trend due to
change in emissions: anthropogenic Swedish (dark blue, Se emis) and full domain, non-
Swedish (fair blue, Eur emis), emissions, trend in top and lateral boundaries of relevant





1 species (yellow, bound) and variation in meteorology (brown, meteo). The sum of the

2 modelled contributions is indicated by the dashed green line.



Table 1a. Model calculations and scenarios, all covering the years 1990-2013, including the
"first guess" to the data assimilation and base case to the sensitivity simulations (MFG), two
reanalysis data sets (LONGTERM and ALL), sensitivity scenarios (MEUR, MSE, MBC and
MMET).

| Scenario/data set | Description |
|---|---|
| MFG | MATCH base case simulation and "first guess" used as input to the reanalyzes. |
| LONGTERM | Reanalysis data set of hourly near-surface ozone concentration covering Sweden and Norway based on 1) the MFG European MATCH simulation and 2) selected hourly near-surface ozone measurements in Sweden and Norway, based on temporal coverage of the measurement sites. Optimal for trend analyses. Analyzed and presented in Sect. 3. |
| ALL | Reanalysis data set of hourly near-surface ozone concentration covering Sweden and Norway based on 1) the MFG European MATCH simulation and 2) all available Swedish hourly ozone measurements and a selection of the Norwegian (as in LONGTERM). Not used for trend analyses in this study, but best estimate for the hourly near-surface ozone concentration in Sweden at any point in time. |
| MEUR | MATCH sensitivity simulation where the full domain anthropogenic emissions are kept constant from year to year, set to the level of 2011. |
| MSE | MATCH sensitivity simulation where the Swedish anthropogenic emissions are kept constant from year to year, set to the level of 2011. |
| MBC | MATCH sensitivity simulation where the top and lateral boundaries for all species are kept constant from year to year, set to the level of 2011. |
| MMET | MATCH sensitivity simulation where the meteorology is kept constant, using the meteorological year 2011. |





1     Table 1b. Formation of contributions to the linear trend over the period 1990-2013 from the

2     sensitivity simulations (Se emis, Eur emis, Bound and Meteo, see Table 1a).

| | |
|---|---|
| Se emis | Contribution to the trend caused by the change in anthropogenic Swedish emissions, calculated as the model scenario difference: MFG-MSE. |
| Eur emis | Contribution to the trend caused by the change in anthropogenic European, non-Swedish, emissions, calculated as the model scenario difference: (MFG-MEUR)-(MFG-MSE). |
| Bound | Contribution to the trend caused by the change in lateral and upper boundaries, calculated as the model scenario difference: MFG-MBC. |
| Meteo | Contribution to the trend caused by the variation in meteorology, calculated as the model scenario difference: MFG-MMET. |
| SUM | Sum of the contributions to the trend, calculated as the sum of: Se emis+Eur emis+Bound+Meteo. |



Table 2. Evaluation of modelled hourly near-surface ozone concentrations in 2013 at Swedish
observation sites. Mean value (mean), standard deviation ($\sigma$), model mean bias normalized by
the observed mean (%bias), Pearson correlation coefficients (r) for data including at least 10
pairs, the root mean square error (RMSE) and number of observed hours at the sites. The
evaluation includes the reanalyzed data sets ALL and LONGTERM, where ALL is evaluated
at the 12 Swedish sites included in that simulation, and LONGTERM is evaluated at the 6
Swedish sites included in that simulation (cf. Fig. 4). For each of these data set evaluations we
include the observation *dependent* reanalysis (2dvar), the observation *independent* cross
validation of the reanalysis (cross) and the MATCH base case simulation (MFG). The top half
of the table shows the temporal performance (spatial mean of statistics, see Supplement Sect.
S1). The bottom half of the table shows spatial performance (spatial statistics of annual
means, see Supplement Sect. S1).

|  |  | spatial mean of hourly statistics | | | | | |
|---|---|---|---|---|---|---|---|
|  |  | mean (ppb(v)) | std dev (ppb(v)) | %bias (%) | r | RMSE (ppb(v)) | #hours |
| ALL | obs | 30.9 | 11.0 |  |  |  | 8760 |
|  | MFG | 31.1 | 9.4 | 1.4 | 0.67 | 8.8 |  |
|  | cross | 30.6 | 9.9 | -0.3 | 0.76 | 8.0 |  |
|  | 2dvar | 30.8 | 11.1 | -0.6 | 0.94 | 3.5 |  |
| LONGTERM | obs | 32.6 | 10.5 |  |  |  | 8760 |
|  | MFG | 31.2 | 9.7 | -3.3 | 0.67 | 8.7 |  |
|  | cross | 32.2 | 9.3 | -0.1 | 0.72 | 8.5 |  |
|  | 2dvar | 32.6 | 10.7 | 0.2 | 0.97 | 2.7 |  |
|  |  | spatial statistics of annual means | | | | | |
|  |  | mean (ppb(v)) | std dev (ppb(v)) | %bias (%) | r | RMSE (ppb(v)) | #stns |
| ALL | obs | 30.9 | 2.5 |  |  |  | 12 |
|  | MFG | 31.1 | 1.2 | 0.6 | 0.21 | 3.0 |  |
|  | cross | 30.6 | 1.8 | -1.0 | 0.11 | 3.5 |  |
|  | 2dvar | 30.8 | 2.8 | -0.5 | 0.98 | 0.7 |  |
| LONGTERM | obs | 32.6 | 2.2 |  |  |  | 6 |
|  | MFG | 31.2 | 1.0 | -4.1 | X | 3.4 |  |
|  | cross | 32.2 | 1.6 | -1.2 | X | 4.3 |  |
|  | 2dvar | 32.6 | 2.2 | 0.2 | X | 0.2 |  |





1    Table 3. Linear trend during 1990-2013 of policy related metrics in the 3 Swedish regions

2    North, Central and South (cf. Fig. 3). Stars (*, **, and ***) indicate that the trend is

3    significant (p≤0.05, p≤0.01, p≤0.001, respectively).

| Metrics | North | Central | South |
|---|---|---|---|
| Mean [$\mu g\ m^{-3}\ year^{-1}$] | **+0.18\*** | +0.13 | **+0.18\*** |
| SOMO35 [ppb(v) d $year^{-1}$] | +14 | -3.1 | -4.7 |
| Maximum 8h mean [$\mu g\ m^{-3}\ year^{-1}$] | -0.11 | **-0.68\*\*** | **-1.2\*\*** |
| Maximum 1h mean [$\mu g\ m^{-3}\ year^{-1}$] | -0.14 | **-0.82\*\*** | **-1.4\*\*\*** |
| AOT40c [ppm(v) h $year^{-1}$] | -0.01 | **-0.07\*** | -0.09 |
| AOT40f [ppm(v) h $year^{-1}$] | +0.03 | -0.09 | **-0.12\*** |
| #hours >80 $\mu g\ m^{-3}$ [# $year^{-1}$] | **+26\*** | +1.7 | +6.6 |
| #days >70 $\mu g\ m^{-3}$ [# $year^{-1}$] | +1.3 | +0.73 | +1.1 |
| #days >120 $\mu g\ m^{-3}$ [# $year^{-1}$] | +0.01 | **-0.12\*** | **-0.32\*\*** |

