# Peer review of "Reanalysis of and attribution to near-surface ozone"

_Atmospheric Chemistry and Physics, 2017_

## Referee Comment (RC1) · Anonymous Referee #1 · 3 Aug 2017

This is an interesting and insightful read and well-written paper that creates two different model re-analyses datasets for ozone, and presents results for annual-mean and annual 1-hour maxima O3 concentrations and trends focussing on three regions in Sweden. Results are presented both seasonally (regional average) and spatially. Trends for different percentiles across the whole distribution as well as vegetation and health metrics are also investigated with interesting differences between North compared to Central and South regions in Sweden. The authors perform sensitivity simulations to attribute these trends and highlight the role of emissions, the O3 hemispheric background and meteorology to both monthly mean O3 and across different parts of the O3 distribution. It does highlight that care is needed when including surface obser-

vations, as these can be influenced by local processes that cannot be captured by the underlying model such as night-time inversions. Overall, there are good insights, but a few clarifications in places would be beneficial.

Major points: 1. The attribution results for the "bound" and "meteo" simulations. The text in this section needs to be clarified, and it would be highly beneficial to establish the sensitivity of the results to the methodology/underlying assumptions. In section 2.5 the method for attributing trends to meteorology and other factors is given as: "The respective contributions to the trend are formed by subtracting the MFG with the corresponding sensitivity simulation" . How exactly how the contributions are calculated i.e is the difference in O3 between the MFG and the sensitivity simulations first calculated to produce a $\Delta$O3 and then trend for $\Delta$O3 or the residual O3 then calculated? Is this trend in $\Delta$ O3 assumed to be the contribution to the O3 trend as plotted in Figs 9 and 10? Please be explicit about this calculation in the text.

A key question is how does this methodology and assumptions made influence the results? in particular:

1) In particular how sensitive are the results in all sensitivity experiments to the choice of the year 2011?

2) For the "bound" simulations, is the year for constant boundary conditions not important, given the assumption of a constant background of O3 from 2000 onwards? The reader also needs to understand further about the O3 boundary conditions: the source of these boundary conditions and why they are assumed constant after 2000. Would there be a much larger contribution to the O3 trend if the boundary conditions varied after 2000?

3) The authors state that the impact of meteorology is difficult to interpret and note meteorology variations cause a positive trend on O3. This does seems rather unintuitive- why is there a positive trend due to meteorology, is this because of the methodology, rather than a robust finding? Varying meteorology is usually noted as the cause of difficulties in O3 trend detection (e.g Colette et al. 2016; Lefohn et al. 2017). Hence there may possibly be a larger O3 trend in a simulation with constant compared to varying meteorology, leaving a residual positive trend? Alternatively, there could be a trend in a given meteorological variable over this period that would cause a trend in O3. It would be highly beneficial to investigate the above points to establish why this trend is positive, and as noted in 1) the sensitivity of this result to the constant meteorological year selected should be assessed for any robust statement to be made.

Colette A, Aas W, Banin L, Braban CF, Ferm, M, et al. 2016. Air pollution trends in the EMEP region between 1990 and 2012. Joint Report of the EMEP Task Force on Measurements and Modelling (TFMM), Chemical Co-ordinating Centre (CCC), Meteorological Synthesizing Centre-East (MSC-E), Meteorological Synthesizing Centre-West (MSC-W). EMEP: TFMM/CCC/MSC-E/MSC-W Trend Report (01/2016)

Lofehn et al (2017) – reference is in the text already.

2. The regional-average trends. How sensitive are these trend results to when the averaging is performed in the calculation, especially in the case of calculating trends in O3 percentile ranges? In this study, regional averaging is done after calculating percentiles at each grid box and then a regional trend is calculated, but trends could be calculated for each grid box first and then averages calculated subsequently or the data could be pooled.

3. Spring and summer peaks. A change in when peak O3 occurs throughout the 24-year period is commented on in several places in the text. However, as noted in Fig S6 there is large inter- interannual variability, such that it is hard to make any robust conclusions about shifts in maxima and dominance of spring vs. summer-time peaks over the course of the trend period. Is there any further evidence to illustrate this point?

4. The data assimilation process. In Figure 1 it seem that the match model simulation feeds into data assimilation, whilst I thought it would be a combination that produces a new simulation. Data assimilation is a process that combines a "background field"

or "first guess" with observations to produces new physically consistent model fields. This is usually an iterative process that occurs as the model runs forward. However, the figure makes this appears as a post –processing correction, although section 2.4 discusses a "first guess". If the process is iterative, the figure does not capture this flow and should be revised for clarity. If this is a post-processing effort the text should be revised to state this.

5. Figure 5 discuses trends in O3 percentile levels. Which percentile levels?

Minor or technical points:

Page 1, Line 13: As above. The second sentence is confusing. I assume the observations are assimilated into the CTM before performing the model simulations.

Page 1, line 20: Please clarify why including all observations leads to artificial trends and why using only time consistent measurements avoid this? Or else remove the text on artificial trends from the abstract for simplicity. See also below. This text appears in a number of places in the manuscript without clarification.

Page 1, line 20: add "Distribution of the" before "surface O3"

Page 1, line 26: change "processes" to something like "factors".

Page 2, line 11: change the IPCC (2013) reference to reference the specific IPCC chapter.

Page2, line 17, is the clause "the right weather conditions" needed? If so why? Also noted below.

Page 4, line 8; Page 5 line 13, page 6 line 12, page 7 line 30, page 10 line 30- in all these places it is highlighted that artificial trends can be introduced, this seems an important and challenging point, can this text be expanded upon so the reader understand why an artificial trend could appear. The text on page 7 is expanded but still not clear. The text on page 10 seems useful earlier. Some of the repetition could

[Figure]

be removed.

Page 18, line 13, as noted above there is large interannual variability that it is hard to say anything about shifts in maxima and dominance of spring vs. summer-time peaks.

Page 8, line 25, As comment 1. above. Please explain the basis of the scaling used and depicted in fig 2a. i.e. what observed changes? Why has this been assumed to be constant from year 2007 onwards? The basis of the seasonal cycle (Fig S1) should also be given.

Page 9, line 10: it would be useful for the reader to provide some insights into the impacts of using a higher resolution emissions data at 1km by 1km over Sweden as compared to EMEP 50 km by 50km when the data are subsequently interpolated to 44km resolution.

Page 9, line 20, The text on the Swedish contribution of emissions to the domain would benefit from a map of the domain, otherwise is this text needed as the methods are already long?

Page 10, line 4: Do these measurement sites have a station classification in EMEP such as "rural", if so please add, so the reader can see that these represent the regional background.

Page 16, line 20 and Figure 6: Please explain how the regional average trends were calculated. Is the regional average calculated first then the trend?

Page 17, line 3: Fig7a shows only two colours so the highest values in the northerly mountains are hard to discern. Could the scale be improved?

Page 17, line 19: Although there are some similarities in spatial patterns the colour scales suggest comparable magnitudes in the south and westernmost part of the region shown for the annual maximum 1 hr mean, whilst the south is the area with highest values for the annual mean metric.

Page 17, line 30 should "in the North" be added after March monthly mean?

Page 19, line 20, please explain what weather states favourable for O3 formation means and why relevant for NOx-VOC regimes for O3 production?

Page 20, line 9, there is also greater photolysis of NO2 to form O3 which is why many locations in the northern hemisphere have a peak in Boreal summer. So greater production as well as destruction.

Page 22, line 18: SOMO35 is not primarily used an indicator of long-term health effects. It is used as a metric compatible with short-term exposure and a threshold for adverse effects to occur- see HRAPIE report by WHO (2013).

WHO 2013b. Health risks of air pollution in Europe – HRAPIE project: New emerging risks to health from air pollution – results from the survey of experts. http://www.euro.who.int/__data/assets/pdf_file/0006/238956/Health-risks-of-air-pollution-in-Europe-HRAPIE-project,-Recommendations-for-concentrationresponse-functions-for-costbenefit-analysis-of-particulate-matter,-ozone-and-nitrogen-dioxide.pdf

Page 23, line 26: data assimilation only reduces the impacts of boundary conditions at the surface.

Page 24, line 11: although a number of studies do find that emissions changes are larger than climate change the time period of the climate change and the metric being analysed is important. Whilst this may be the case for annual or summertime means for higher frequency metrics such as percentiles used here this has not been well established.

Page 25, line 8, SOMO35 in the north part of Sweden increase, hence this conclusion should be modified.

Figure 1: see comment 4 above.

[Figure]

Figure 5: See comment 5 above. This figure describes trends in percentile levels but please include the percentile levels that are plotted.

Figure 8-It is hard to see any trend in this figure because of the scale that covers all the percentile ranges. Could separate panels for low –median and median to high percentiles help with clarity? Or perhaps a figure like Fig 5 in Simpson et al. 2014 which depicts the table results clearly. An alternative would be to swap Figure 8 to the supplement and Table S3 to the main text as the table shows the trend results more clearly than the Figure.

Figure 9- the legend is hard to read.

Figure S6- this figure displays two y-axes but only one set of points are plotted?

[Figure]

---

## Referee Comment (RC2) · Anonymous Referee #2 · 7 Aug 2017

This study is a nice example of combining models with observations, and making use of the model to better understand the causes of observed trends. In general the paper is nicely written, and will deserve publication in ACP after attention to the following points.

I have two significant reservations:

1. As noted on page 11, the length scale used for the 2dvar is 1000 km. The authors note that this is large, but claim that this is justified by the sparse network and the weak gradients in Sweden. However, Fig. 7 makes clear that the gradients can be rather large, especially in Southern Sweden. I would like to see more discussion of

this problem, ideally with results from some sensitivity runs to help demonstrate if this really is a serious issue or not.

2. Much of the discussion around annual mean O3 values results from the problem of nocturnal ozone depletion, which is said to be more important in southern Sweden. As nocturnal O3 itself is quite irrelevant for most health and vegetation metrics, why wasn't the analysis focused on some ozone-indicator that actually reflects these problems? This could be daytime ozone, M7, M12, or the daily 8-h values mentioned in Table 3.

Page-by-page comments:
* * *
Page 1, Abstract

l16 - ... 'performance over' rather than 'performance than'

l9-11 - use more recent refs.

l14 - define NOx as NO + NO2.

l21-22 - the paper of Fiore et al (2011) provides a much more recent example of this PAN effect.

l31 - here the HTAP results presented in Fiore et al (2009) would also be relevant.

l16 - use more recent refs

l13 - move technique before ()

l30 - explain or provide references for 'databases EMEP and Airbase'

l22-23 - states that MATCH only calculates chemistry for the lowest 5 km of the model domain. Is this a sufficient depth, when looking at the impact of tropospheric boundary conditions?

I was puzzled as to why the boundary conditions are given as mass units. Usually the volumetric mixing ratio is used as this is the more conserved quantity, and independent of pressure and temperature. Why this choice?

l18 - states that the model uses 22km grid-spacing, but on page 9 the emissions are interpolated to a 44km grid. Which is correct?

l18 - states that the model uses 22km grid-spacing, but on page 9 the emissions are interpolated to a 44km grid. Which is correct?

l26 - maybe add 'see also Andersson et al, 2007' here as a ref also, since it isn't obvious from the text where the time-development comes from. Further confusion arises on page 23, when it is stated that the trends are taken from Engardt et al., 2017. Please clarify which statement is correct?

l25-26 - states that no trend is assumed in the intra-annual variation in emissions, but such a trend is likely to exist. There have been quite large changes in the sources and fuel-mix over this period. Will this matter? I think you should also mention that the spatial distribution of emissions is also held constant, which is possibly a bigger source of uncertainty. (Which year was used for the spatial distribution?)

Page 10. Are these sites all part of EMEP? If not, are the data-quality criteria equivalent to those of EMEP sites?

What is 'full-domain'? Is all of Europe covered? Does 'Eur' include Russia?

Page 12 and onwards The 2-letter code 'Se' is a little confusing, neither English nor an accepted abbreviation. The UN code is 'SE', so why not use that directly?

l1 - mention that numerics can also cause non-linearity in CTMs.

l26 - this says Fig. 5 gives 'annual' O3, but Fig.5 mentions just 'percentiles'. Which percentiles? I couldn't figure out what this Figure was showing.

l14 - why say 'possibly caused by'? You have the data, so you can say exactly what caused this.

l21-22. This explanation of Fig.8 would have been better presented before it is first referenced

l14-15. This Figure reminds strongly of that presented by Jenkin (2008), so it would be good to reference that paper.

l22-26. The big change in sign for 'meteo' between the 98th and 100th percentile deserves some comment.

l23-26. This sounds like a political statement of the authors views. I agree that NOx control is essential for many reasons, but cite scientific papers to support your statement.

l21-22. This statement is unclear. Which earlier studies?

Figures

————-

Generally, the figure quality is quite poor and should be improved. (Some of the figures look like screen-dumps of excel plots, and the various Sweden maps (e.g. S10-S11) have awful color schemes.)

Fig. 2. The $C_5H_8$ emissions are so close to zero here that the plot doesn't show anything except that the emissions are very small. These could either be presented on a separate plot, or just described in the text.

Are $C_5H_8$ emissions really so small by the way? I have seen larger estimates for Europe.

Fig. 3. Given the frequent discussion of the topographic location of these sites, I think a Table with altitude would also help.

Fig. 4. I found the color choice unusual. Usually one uses red to indicate a warning, e.g. that data-quality is poor. Here red is used to indicate good data-quality,

Fig. 5. As noted above, I don't know what 'ozone percentiles' means if one doesn't specify which percentile. The blue and green colors here can also be hard to distinguish.

Fig. 7. Poor quality.

Fig. 8. Increase the font-size for the percentile labels - they are really hard to see.

Fig. 9. Improve quality. I really liked the content of this Figure, and also Fig. 10, but they both look like screen dumps.

**References**

——————-

Jenkin, ME, Trends in ozone concentration distributions in the UK since 1990: Local, regional and global influences, Atmos. Environ., 42, 5434-5445, 2008

Fiore, AM., Levy II, H. & Jaffe, D., A. North American isoprene influence on intercontinental ozone pollution, Atmos. Chem. Physics, 11, 1697-1710, 2011

Fiore, A., Dentener, F., Wild, O., et al., A., Multi-model estimates of intercontinental source-receptor relationships for ozone pollution, J. Geophys. Res., 114, 2009

---

## Author Comment (AC1) · 21 Sep 2017

We wish to thank the referee for insightful comments and questions, as well as technical improvements. The study has much benefited from the additional work performed as a consequence of the major points. Point by point replies follow below to all issues raised by the referee. We also attach the manuscript (including figures and tables) and supplements with track changes to clearly show which changes have been made. Please observe that this also includes changes/updates based on anonymous referee #2.

Major points:

[Figure]

1. The attribution results for the "bound" and "meteo" simulations. The text in this section needs to be clarified, and it would be highly beneficial to establish the sensitivity of the results to the methodology/underlying assumptions. In section 2.5 the method for attributing trends to meteorology and other factors is given as: "The respective contributions to the trend are formed by subtracting the MFG with the corresponding sensitivity simulation" . How exactly how the contributions are calculated i.e is the difference in O3 between the MFG and the sensitivity simulations first calculated to produce a _O3 and then trend for _O3 or the residual O3 then calculated? Is this trend in _ O3 assumed to be the contribution to the O3 trend as plotted in Figs 9 and 10? Please be explicit about this calculation in the text.

Reply from the Authors (RA): The contributions the trend is calculated by 1. Forming gridded metrics (annual/monthly), 2. Gridded trends, 3. Regional average trend, 4. Difference between MFG and the corresponding sensitivity simulation. We have clarified this in the method description Sect. 2.5:

"The respective contributions presented in Sect. 3.3 are formed through the following sequence: 1. Calculation of gridded metrics (focusing on monthly 1h maximum, monthly mean and annual 1h percentile levels); 2. Calculation of secular gridded trends over the monthly or annual metrics; 3. Calculation of regional (North, Central, South, cf. Fig. 3) mean of the secular trends, 4. Calculation of the difference between the regional mean trend in MFG and the corresponding sensitivity simulation."

A key question is how does this methodology and assumptions made influence the results? in particular: 1a. In particular how sensitive are the results in all sensitivity experiments to the choice of the year 2011?

RA: This is indeed an important question.

It is too much for us to test all 24 years as base year. To restrict the amount of CPU-efforts, we have made a sensitivity test where we use 1990 as base year instead for bound, meteo and emis (where emis SE and emis Eur are combined, this contribution

is included in Table 1b). 1990 was chosen as it differs from 2011 both for European emissions and climatologically. For example the NAO index was high in early winter 1990 and low in 2011, whereas the summer index was positive but close to 0 in 1990 and negative in 2011.

We choose to show the differences in contributions to the trend due to base year by scatterplots in Fig 5c and the Supplements (Fig. S5). They show that the contributions are robust for percentiles and most monthly means. Whereas the contributions of variations in meteorology and emissions to the secular trend in monthly max 1h mean differ for some months.

We have included a description on how we conduct this investigation in Sect. 2.5 and 2.6., and present the results in Sect. 3.1.:

"Third, the attribution may be sensitive to the chosen base year. Sensitivity simulations using 1990 as base year instead of 2011 are also conducted, to investigate the robustness of the results. To investigate all 24 years as base years would take too much computational efforts, we choose 1990 as it differs from 2011 both for European emissions and climatologically. If the contributions to the trend differ too much between the base years 1990 and 2011 then the results are not robust. If they are similar it is not a guarantee that the results are robust but it is an indication. The contributions with 1990 as base year are formed in the same way as for the 2011 sensitivity runs. The contributions due to change in top and lateral boundaries (bound) and variations in meteorology are included in the same manner, while we compare the total footprint of the change in emissions, i.e. the sum of FD emis and SE emis (emis) rather than the two parts."

"Finally, investigating the impact of the selected base year in the sensitivity simulations, we compare the contributions (bound, emis and meteo) based on keeping the year 2011 constant in the sensitivity simulations to keeping the year 1990 constant (Fig. 5c). Most contributions to the trend in percentiles are robust (Fig. 5c), falling close to

the 1:1 line. Only a few of the very weakest O3 contribution trends fall outside the factor 2 lines (for the meteo contribution). The contributions to the secular trend in some of the monthly mean and the monthly maximum 1h mean near-surface O3 differ more for the two base years than the percentiles (Supplements, Fig. S5). For monthly mean the trend due to changes in meteorology is stronger for some months (one month is weaker) when 2011 is used as base year compared to 1990. The other contributions fall within the factor 2 lines. For monthly maximum the deviation is larger, even differing in sign for the contribution due to variation in meteorology for some months, and a few contributions due to emission change also fall outside the factor 2 lines."

1b. For the "bound" simulations, is the year for constant boundary conditions not important, given the assumption of a constant background of O3 from 2000 onwards? The reader also needs to understand further about the O3 boundary conditions: the source of these boundary conditions and why they are assumed constant after 2000. Would there be a much larger contribution to the O3 trend if the boundary conditions varied after 2000?

RA: There is observational evidence that near surface ozone in Europe has increased up to ca. 2000, with less clear trends after that date (e.g. Cooper et al., 2014). We have therefore selected a conservative approach and let all the ozone concentrations at the boundaries be fixed from 2000 onwards .This is now clarified in the text.

In the present model set-up we also do not have the possibilities to assign different trends at different geographical locations or different trends for different ozone metrics (e.g. maximum concentrations or average concentrations). If we would have assumed continued ozone increase after 2000 (in opposition to observations) our results would likely shift towards a larger increase (or smaller decrease) of near surface ozone during the period 1990-2013. It is not possible, at this stage, to discuss exactly how that would impact the different ozone percentiles.

1c. The authors state that the impact of meteorology is difficult to interpret and note

meteorology variations cause a positive trend on O3. This does seems rather unintuitivewhy is there a positive trend due to meteorology, is this because of the methodology, rather than a robust finding? Varying meteorology is usually noted as the cause of difficulties in O3 trend detection (e.g Colette et al. 2016; Lefohn et al. 2017). Hence there may possibly be a larger O3 trend in a simulation with constant compared to varying meteorology, leaving a residual positive trend? Alternatively, there could be a trend in a given meteorological variable over this period that would cause a trend in O3. It would be highly beneficial to investigate the above points to establish why this trend is positive, and as noted in 1) the sensitivity of this result to the constant meteorological year selected should be assessed for any robust statement to be made.

RA: It is indeed a fact that meteorology introduces interannual variations that can cause a trend not to be significant. The trend can also be sensitive to extreme years in the beginning/end of the time series. We have made a few sensitivity tests: i/ using 1990 as base year (see 1b above). ii/ investigating the overall trend over shorter time period periods (1990-2012, 1990-2011, 1991-2013, 1992-2013).

The attached Table X1 (replica of Table 3 for different periods, investigation of point ii) shows that the trend is not sensitive to removal of 1 or two years in the beginning. We choose not to include this sensitivity analysis since the manuscript is long already.

However, we do not agree that it is un-intuitive that the meteorological variations could cause a positive ozone trend in Sweden. Warmer climate could cause more biogenic isoprene emissions in Sweden and elsewhere and more high pressures and transport events of high ozone from the European continent could also contribute. This is in agreement with what was found in the retrospective simulations of Andersson et al. (2007), a positive but not significant trend of ca 0.5% decade due to meteorological variability over the periods 1958-2001 and 1979-2001. It is outside the scope of this study to investigate the reason behind this.

Changes in manuscript: see 1a.

2. The regional-average trends. How sensitive are these trend results to when the averaging is performed in the calculation, especially in the case of calculating trends in O3 percentile ranges? In this study, regional averaging is done after calculating percentiles at each grid box and then a regional trend is calculated, but trends could be calculated for each grid box first and then averages calculated subsequently or the data could be pooled.

RA: We have also calculated the trend per grid box, after calculating the percentiles per grid box (see also Fig. 7 and supplements) and then conducted the spatial averaging to compare this to Table 3. The results are identical, see attached Table X2.

The trends are the same under these two methods as a result of us using first order linear regression. We do not think it is correct to average the p-value over a region, still we include this as well in the below table (same indication as in Table 3). The two methods result in somewhat different significances, which most often are stronger than those of the trend in the regional average (except for annual mean and number of hours above 80 ppbv).

To calculate pooled trend statistics is outside the scope of this study. We do not include these results as the manuscript is long already.

3. Spring and summer peaks. A change in when peak O3 occurs throughout the 24-year period is commented on in several places in the text. However, as noted in Fig S6 there is large inter- interannual variability, such that it is hard to make any robust conclusions about shifts in maxima and dominance of spring vs. summer-time peaks over the course of the trend period. Is there any further evidence to illustrate this point?

RA: as the manuscript is long already we remove the discussion on the shift in peak rather than adding more analyses.

4. The data assimilation process. In Figure 1 it seem that the match model simulation feeds into data assimilation, whilst I thought it would be a combination that produces

a new simulation. Data assimilation is a process that combines a "background field" or "first guess" with observations to produces new physically consistent model fields. This is usually an iterative process that occurs as the model runs forward. However, the figure makes this appears as a post –processing correction, although section 2.4 discusses a "first guess". If the process is iterative, the figure does not capture this flow and should be revised for clarity. If this is a post-processing effort the text should be revised to state this.

RA: The variational data analysis was conducted as a post processing, i.e. in retrospect. Thus the figure is correct.

We have clarified in the text by refraining from calling it data assimilation and instead calling it (retrospective) variational data analysis. We choose to keep the terminology "first guess" of the model background field to the variational data analysis since we think the text is clear enough by the introduction of these changes.

5. Figure 5 discuses trends in O3 percentile levels. Which percentile levels?

RA: We do not expect the reader to understand which circle belongs to which percentile (although this can be achieved by comparison to e.g. Supplement Table S2 or Fig. 10), however the levels included were lacking. We have included the percentile levels in the figure legend and manuscript text.

Minor or technical points:

Page 1, Line 13: As above. The second sentence is confusing. I assume the observations are assimilated into the CTM before performing the model simulations.

AR: The data assimilation was conducted as a post process. We have clarified this by stating "using retrospective variational data analysis".

Page 1, line 20: Please clarify why including all observations leads to artificial trends and why using only time consistent measurements avoid this? Or else remove the text on artificial trends from the abstract for simplicity. See also below. This text appears in

number of places in the manuscript without clarification.

RA: we have removed the statement about "artificial trends" from the abstract. See more on the issue below.

Page 1, line 20: add "Distribution of the" before "surface O3"

AR: Done.

Page 1, line 26: change "processes" to something like "factors".

AR: Done.

Page 2, line 11: change the IPCC (2013) reference to reference the specific IPCC chapter.

AR: Done.

Page2, line 17, is the clause "the right weather conditions" needed? If so why? Also noted below.

RA: Yes. This is to emphasize that the weather does affect ozone formation, e.g. cloudy and/or cold weather affects the ozone formation negatively whereas warm and clear weather means favorable conditions. We have changed the language to "under favorable weather conditions"

Page 4, line 8; Page 5 line 13, page 6 line 12, page 7 line 30, page 10 line 30- in all these places it is highlighted that artificial trends can be introduced, this seems an important and challenging point, can this text be expanded upon so the reader understand why an artificial trend could appear. The text on page 7 is expanded but still not clear. The text on page 10 seems useful earlier. Some of the repetition could be removed.

RA: We have now included an explanation at the first mentioning of artificial trend in the introduction: "Artificial trends can for example arise from the introduction of new observation sites, which reduce the model bias in the area surrounding the measurement site during the time it is included but not before." Thus, if the model bias is prone to be constant over-estimation (or under-estimation) then the introduction of more (or less) observations by time will cause artificial trends in the data set. We have also removed some of the repetition but as it is a vital point in this study we keep most of them.

Page 8, line 25, As comment 1. above. Please explain the basis of the scaling used and depicted in fig 2a. i.e. what observed changes? Why has this been assumed to be constant from year 2007 onwards? The basis of the seasonal cycle (Fig S1) should also be given.

RA: The temporal trend of the boundary concentrations used in the present study is, admittedly, a crude estimate but should grossly represent the evolution of atmospheric species relevant to tropospheric ozone. Trends in $NO_y$ concentrations follow emission changes in the USA, $CH_4$ concentrations closely follow global average background concentrations. $O_3$ concentrations are assumed to increase by 1% per annum until 2000 and held constant after that date. The reason for this is to use a pragmatic and conservative approach considering all involved uncertainties (as introduced in an earlier published study; Engardt et al, 2017).

Fig. S1 illustrates the absolute levels and annual variation of $O_3$ boundary concentrations during the year 2011. The values and their seasonal variation are from MATCH standard configuration as outlined in Andersson et al. (2007).

Andersson, C., Langner, J., and Bergström, R.: Interannual variation and trends in air pollution over Europe due to climate variability during 1958 2001 simulated with a regional CTM coupled to the ERA40 reanalysis, Tellus B 59, 77-98, 2007. Engardt, M., Simpson, D., Schwikowski, M. and Granat, L.: Deposition of sulphur and nitrogen in Europe 1900-2050. Model calculations and comparison to historical observations, Tellus B, 69, 1328945, 2017.

Page 9, line 10: it would be useful for the reader to provide some insights into the

impacts of using a higher resolution emissions data at 1km by 1km over Sweden as compared to EMEP 50 km by 50km when the data are subsequently interpolated to 44km resolution.

RA: The gain is mainly in the interpolation of the 50km resolution emissions to 44km, on a different projection, where we lose spatial information. SMED and EMEP are otherwise very similar for Sweden. This is already stated in the text: "National totals from SMED are very similar to the national totals available in the EMEP database, but our methodology enables higher resolution emission data over Sweden."

The reason for us doing this is also technical. The high-resolution emission inventory is used in later steps in the MATCH Sweden system. This is not explained in this manuscript, but a later, where this issue will be investigated. We wished to use as similar input data as possible also in this study. We ask the referee to wait for the next paper to get a more in-depth analysis.

Page 9, line 20, The text on the Swedish contribution of emissions to the domain would benefit from a map of the domain, otherwise is this text needed as the methods are already long?

RA: Yes. Actually the manuscript will benefit from a map of both domains: the total European domain and the domain of the "2dvar analysis". We have added this to the manuscript (Fig. 1b).

Page 10, line 4: Do these measurement sites have a station classification in EMEP such as "rural", if so please add, so the reader can see that these represent the regional background.

RA. The sites are classified as regional background sites. We have added a more thorough description of the origin and scale the sites represent.

Page 16, line 20 and Figure 6: Please explain how the regional average trends were calculated. Is the regional average calculated first then the trend?

RA: Here we averaged over the region before computing the trend. We have clarified this in the figure legend and in the main text.

Page 17, line 3: Fig7a shows only two colours so the highest values in the northerly mountains are hard to discern. Could the scale be improved?

RA: Of course. We have changed the scale and improved the resolution of the figures.

Page 17, line 19: Although there are some similarities in spatial patterns the colour scales suggest comparable magnitudes in the south and westernmost part of the region shown for the annual maximum 1 hr mean, whilst the south is the area with highest values for the annual mean metric.

RA: we have clarified the sentence: "...have similar spatial variation as their respective period means (Fig. 7), i.e. the maximum of annual means peaks in the north and the maximum of the annual maximum 1h mean peaks in the south"

Page 17, line 30 should "in the North" be added after March monthly mean?

RA: we have added "in the Central and North".

Page 18, line 13, as noted above there is large interannual variability that it is hard to say anything about shifts in maxima and dominance of spring vs. summer-time peaks.

RA: we have removed this since the manuscript is long already (major point 3).

Page 19, line 20, please explain what weather states favourable for O3 formation means and why relevant for NOx-VOC regimes for O3 production?

RA: We apologize for an unfortunate typo; the text should have been "... not favorable for O3 formation". We have now reformulated to emphasize when this effect is the strongest: "when there is little photolysis (especially in the winter and during night-time)"

Page 20, line 9, there is also greater photolysis of NO2 to form O3 which is why many

locations in the northern hemisphere have a peak in Boreal summer. So greater production as well as destruction.

RA: The discussion deals with the fact that the trend in the boundaries contribution to the trend in near-surface concentrations over Sweden is lower during summer than during rest of the year despite the trend in boundary forcing being the same throughout the year in this numerical experiment. Our conclusion is that this is related to processes responsible for decreasing the life time of near surface ozone. We listed two such processes "dry deposition to vegetation and photolysis of ozone". We agree that O3 production is also greater in summer, but it is not relevant to include it in that paragraph.

Page 22, line 18: SOMO35 is not primarily used an indicator of long-term health effects. It is used as a metric compatible with short-term exposure and a threshold for adverse effects to occur- see HRAPIE report by WHO (2013).

RA: Yes. This was a mistake. As the paragraph was too political (see referee #2) and not completely correct as you point out we choose to remove the whole lot. We define SOMO35 including references that we think are well suited earlier in the section.

WHO 2013b. Health risks of air pollution in Europe – HRAPIE project: New emerging risks to health from air pollution – results from the survey of experts. http://www.euro.who.int/__data/assets/pdf_file/0006/238956/Health-risks-of-airpollution- in-Europe-HRAPIE-project,-Recommendations-for-concentrationresponsefunctions- for-costbenefit-analysis-of-particulate-matter,-ozone-and-nitrogendioxide. pdf

Page 23, line 26: data assimilation only reduces the impacts of boundary conditions at the surface.

RA: we now clarify that we mean "reduces the impact at the surface caused by errors in the lateral and upper model boundaries".

Page 24, line 11: although a number of studies do find that emissions changes are

larger than climate change the time period of the climate change and the metric being analysed is important. Whilst this may be the case for annual or summertime means for higher frequency metrics such as percentiles used here this has not been well established.

RA: Thank you for pointing this out. We have clarified that the statement is meant for annual and summertime means. Further we include a new paragraph in the discussion:

"Our study shows that the impact of meteorological variability on the trend changes strongly from lower percentile levels to the very highest (in the South), with a shift from a positive to a negative contribution (cf. Fig. 10). Thus, conclusions drawn on the importance of meteorological variability in comparison to other factors such as changes in emissions will vary strongly depending on the metric that is studied. We have also studied the impact of base year in the sensitivity study (1990 vs 2011; cf. Fig. 5c and Supplement Fig. S5). The attribution to the trend is robust for all percentiles, including the annual maximum, whereas the monthly maximum is not robust for emissions and meteorological variation. So far studies of the future development of near-surface O3 have focused on long-term means such as summer mean (e.g. Langner et al, 2012a,b; Watson et al., 2016), whereas the the direction of cause of high-frequency metrics, such as the higher percentiles we show here, have not been established and should be investigated further."

And also as a new bullet in the conclusions.

Page 25, line 8, SOMO35 in the north part of Sweden increase, hence this conclusion should be modified.

RA: Yes. We have clarified that this is true for central and south. In the north the changes are not significant so we refrain from conclusions on that.

Figure 1: see comment 4 above.

RA: The figure is correct.

Figure 5: See comment 5 above. This figure describes trends in percentile levels but please include the percentile levels that are plotted.

RA: the percentile levels are included in the main text as well as in the legend. See also reply above.

Figure 8-It is hard to see any trend in this figure because of the scale that covers all the percentile ranges. Could separate panels for low –median and median to high percentiles help with clarity? Or perhaps a figure like Fig 5 in Simpson et al. 2014 which depicts the table results clearly. An alternative would be to swap Figure 8 to the supplement and Table S3 to the main text as the table shows the trend results more clearly than the Figure.

RA: We have redrawn the figure to increase the legibility. We prefer to retain the figure with its three panels as it gives a comprehensive graphical overview of the trends of the different percentiles in the different parts of Sweden. The exact numbers of the trends and their significance, provided in the Supplement, are not the main message, but is available in other places.

Figure 9- the legend is hard to read.

RA: ok, we have updated the resolution of the figure.

Figure S6- this figure displays two y-axes but only one set of points are plotted?

RA: The left hand axis shows the running day of the year and the right hand axis the date this means. We remove the figure, see above.

Please also note the supplement to this comment:
https://www.atmos-chem-phys-discuss.net/acp-2017-338/acp-2017-338-AC1-supplement.zip

---

## Author Comment (AC2) · 21 Sep 2017

We would like to thank the anonymous referee #2 for insightful comments and questions. We appreciate the raising of the two significant reservations, by addressing these we feel the manuscript has become more focused but also more nuanced. The technical issues has also helped us to improve the manuscript greatly. We have attached (as a supplement to this reply) the manuscript and supplements with track changes included to allow for an overview of all changes included. Please note that this includes all changes to the manuscript, also including those based on anonymous referee #1. Below follow our replies point by point to the issues raised by the referee.

[Figure]

I have two significant reservations:

1.As noted on page 11, the length scale used for the 2dvar is 1000 km. The authors note that this is large, but claim that this is justified by the sparse network and the weak gradients in Sweden. However, Fig. 7 makes clear that the gradients can be rather large, especially in Southern Sweden. I would like to see more discussion of this problem, ideally with results from some sensitivity runs to help demonstrate if this really is a serious issue or not.

Reply from the Authors (RA): Unfortunately we cannot perform any new sensitivity runs testing the length scale due to technical reasons. We will have to leave this for future work.

We agree that the length scale may be too large in the southern part of the country. The impact of this may be too weak gradients in the south in the reanalysis. We have added a discussion on this in the in Sect. 4, whilst also shifting parts of the text from the methods to the discussion to focus the discussion in one place:

"The length scale of the variational data analysis is set to 1000 km, implying a large horizontal influence of the observation increments. This is related to the sparse network of regional background observations but also the relatively small emissions of O3 precursors in Sweden resulting in weak horizontal gradients of near-surface O3 on the regional background scale. The large length scale is also a filtering of local influences in the observations, consequently suppressing sharp gradients in the analysis. However, the horizontal variation in near-surface ozone is larger in the south than in the north, and the large length scale chosen in the data analysis may cause too weak horizontal gradients in the reanalysis data set, especially in the south. An improvement to this would be to describe the geographical variation of near-surface ozone in the background error field, rather than representing this by a constant value as done in this study."

2. Much of the discussion around annual mean O3 values results from the problem of

nocturnal ozone depletion, which is said to be more important in southern Sweden. As nocturnal O3 itself is quite irrelevant for most health and vegetation metrics, why wasn't the analysis focused on some ozone-indicator that actually reflects these problems? This could be daytime ozone, M7, M12, or the daily 8-h values mentioned in Table 3.

RA: We have included also an evaluation of the daily maximum 1h mean ozone in 2013 (Table 2). We choose the daily maximum rather than other metrics, since the annual/monthly maximum is one of the main focuses of the manuscript. The daily maximum is highly likely to occur during daytime, meaning it should not be impacted by the nocturnal ozone depletion. The evaluation shows overall improved scores compared to the hourly/annual mean. However, similar to the annual mean evaluation, the spatial correlation is 0.1 units worse in the cross validation as compared to the MFG simulation. Our conclusion is therefore that it is not only the nightly inversions that cause problems in the spatial variation, but this also occurs during daytime (for the highest values). This can be caused for instance by the distance to oceans and emission sources. We have included the evaluation (in Sect 3.1) and a discussion on this topic, also reducing the focus on the nocturnal ozone depletion (night-time inversions) in the manuscript: "The evaluation of the daily maximum generally shows better correlation but slightly larger bias than the evaluation of the hourly mean. The spatial correlation is also worse in the cross validation compared to the MFG, but the spatial error is improved." We removed the discussion on the impacts of the local topography in Sect. 3.1 (evaluation) and focus this to the discussion in Sect. 4.

Page-by-page comments:

Page 1, Abstract

l16 - ... 'performance over' rather than 'performance than'

RA: Done

l9-11 - use more recent refs.

RA: The introduction is opened by general and non-controversial statements. We feel the cited overview literature is relevant although they may be old according to some standards.

l14 - define NOx as NO + NO2.

RA: Done. We also moved the definition of NO and NO2 to the Introduction where NOx is defined.

l21-22 - the paper of Fiore et al (2011) provides a much more recent example of this PAN effect.

RA: a reference to Fiore et al 2011 is added.

l31 - here the HTAP results presented in Fiore et al (2009) would also be relevant.

RA: a reference to Fiore et al. 2009 is added.

l16 - use more recent refs

RA: We agree the references are outdated. To simplify we changed the sentence and references to: "The strong increase in near-surface O3 concentration until the late 1990s at Mace Head, has levelled out to relatively stationary annual values throughout the 2000s (Derwent et al., 2013; Cooper et al., 2014)."

l13 - move technique before ()

RA: Done

l30 - explain or provide references for 'databases EMEP and Airbase'

RA: We found an error in this sentence. EMEP is now referred to later in the

manuscript, whereas Airbase is not included anymore. The text on this MACC re-analysis was updated and we shifted its location in the introduction:

"Another reanalysis of near-surface O3 concentration in Europe, also within the MACC project, was conducted for the period 2003-2012 (Katragkou et al., 2015). In this re-analysis 4dvar data assimilation was also used to incorporate retrievals from satellites. The data assimilation reduced the bias in near-surface O3 concentration in most of Europe, and it reproduced the summertime maximum in most parts of Europe, but not the early spring peak in northern Europe."

l22-23 - states that MATCH only calculates chemistry for the lowest 5 km of the model domain. Is this a sufficient depth, when looking at the impact of tropospheric boundary conditions? I was puzzled as to why the boundary conditions are given as mass units. Usually the volumetric mixing ratio is used as this is the more conserved quantity, and independent of pressure and temperature. Why this choice?

RA: MATCH has been constructed to describe near-surface concentrations and surface depositions. MATCH does not include stratospheric chemistry. In its standard European configuration we therefore typically limit the vertical domain to the lowest 5-6 km of the troposphere. Through numerous comparisons with observations and other similar models (see references in the main text) we have great confidence in the model's ability to reproduce near-surface O3 in Europe. We believe the present set-up is adequate for assessments of the impact of general trends in hemispheric background concentrations although trends in stratospheric chemistry or physically driven changes in stratospheric-tropospheric exchanges will likely not be captured. This comment is now also introduced to the main text.

In the original version we chose to present mass concentrations in order to use consistent units throughout the manuscript, but agree it is better to show the conserved quantity that is actually used by the model. We have now updated Figure S1 to display

boundary concentrations in ppb(v) (MATCH uses g/kg).

l18 - states that the model uses 22km grid-spacing, but on page 9 the emissions are interpolated to a 44km grid. Which is correct?

RA: This statement of resolution refers to that of the original HIRLAM meteorology. The resolution of MATCH (which we also interpolate the meteorology to) is given previously in section 2.2, as we wish to explain the meteorology separately to the MATCH setup. We added "and emissions" to the sentence in section 2.2.

The reason for running MATCH on a coarser resolution than that of our Swedish emissions and the original meteorology is the need to limit CPU resources consumed by the project.

l26 - maybe add 'see also Andersson et al, 2007' here as a ref also, since it isn't obvious from the text where the time-development comes from. Further confusion arises on page 23, when it is stated that the trends are taken from Engardt et al., 2017. Please clarify which statement is correct?

RA: Sorry for the confusion. We have amended the text and now refer to Engardt et al. (2017) already in section 2.2.1.

The secular trends are from Engardt et al. (2017), while the reference boundary concentrations -valid for the year 2000- are taken from Andersson et al. (2007). As explained in Andersson et al. (2007) do we include seasonal and geographical variations of the boundary concentrations in the reference case.

l25-26 - states that no trend is assumed in the intra-annual variation in emissions, but such a trend is likely to exist. There have been quite large changes in the sources and fuel-mix over this period. Will this matter? I think you should also mention that the

spatial distribution of emissions is also held constant, which is possibly a bigger source of uncertainty. (Which year was used for the spatial distribution?)

RA: Both the spatial distribution and the seasonal varia- tion is updated by the year year, e.g. for SMED see http://extra.lansstyrelsen.se/rus/SiteCollectionDocuments/Statistik%20och%20data/Nationell%20emissionsdatabas/Metod (in Swedish). We have removed the incorrect statement from the manuscript.

Page 10. Are these sites all part of EMEP? If not, are the data-quality criteria equivalent to those of EMEP sites?

RA: We have added a detailed description of all utilized sites in the main text:

"The Swedish observations were delivered by the Swedish data host (at that time, July 1, 2017, Swedish Environmental Institute, IVL). The Norwegian observations were ex- tracted from EBAS (http://ebas.nilu.no; extracted on July 6, 2017). All sites except Norr Malma and Rödeby are classified as regional background measurement sites by EMEP (Internet URL: http://www.nilu.no/projects/ccc/emepdata.html; Hjellbrekke and Solberg, 2015). Norr Malma is located ca 70 km north-east of Stockholm and is considered a regional background measurement site by Stockholm Air and Noice (http://slb.nu), who are responsible for the site. Rödeby is located 10 km north of the small town Karlskrona, and is considered a rural location (personal communication with Titus Kyrklund, Swedish EPA)."

What is 'full-domain'? Is all of Europe covered? Does 'Eur' include Russia?

RA: We have included a map showing the full domain in Fig. 1 (panel b). We refer to this in the bullet. We have also re-defined and explained the labels of the two bound simulations: "SE emis" is Swedish only emissions (see below)," FD emis" is full domain minus Swedish emissions.

Page 12 and onwards The 2-letter code 'Se' is a little confusing, neither English nor an

accepted abbreviation. The UN code is 'SE', so why not use that directly?

RA: We have changed the code to SE emis from Se emis.

l1 - mention that numerics can also cause non-linearity in CTMs.

RA: yes. We added this to the end of the sentence: ", or as a numeric effect in the model."

l26 - this says Fig. 5 gives 'annual' O3, but Fig.5 mentions just 'percentiles'. Which percentiles? I couldn't figure out what this Figure was showing.

RA: The figure shows whether the trends averaged over the three regions for different percentile levels are the same between the SUM, MFG and LONGTERM data sets (values for the latter are also given in Supplementary Table S3). The issue is not to specifically understand which circle that belongs to which percentile, however the information on the levels used was lacking.

To clarify, we have changed the description in the description in Sect 3.1 to "In Fig. 5 we compare regionally averaged linear trends in annual percentiles (levels: 0, 2, 5, 10, 25, 50, 75, 90, 95, 98 and 100) of hourly near-surface O3 over the period 1990-2013. . ." . We have also updated the figure legend.

The figure was also updated with a panel c due to a comment by Referee #1.

l14 - why say 'possibly caused by'? You have the data, so you can say exactly what caused this.

RA: We have checked the data. Both daytime and nighttime means are lower in the south than in the northerly mountains although the difference is larger for the nighttime mean. Thus we conclude that it is mainly an impact of the higher altitude in the northerly mountains that cause higher mean values in the north, while in the south the lower values are partly caused by the lower nighttime values and the higher max-values are caused by high ozone events originating mainly from continental Europe.

We have edited the text accordingly: "The lower period mean of the near-surface O3 in the south than in the north is mainly caused by the higher altitude, of the latter, mountainous, region whereas the opposite gradient for the annual maximum 1h mean is caused by the distance to continental Europe where the high ozone events originate from."

l21-22. This explanation of Fig.8 would have been better presented before it is first referenced

RA: We have moved the explanation to the first paragraph in Sect. 3.2.

l14-15. This Figure reminds strongly of that presented by Jenkin (2008), so it would be good to reference that paper.

RA: The paper is now included as a reference.

l22-26. The big change in sign for 'meteo' between the 98th and 100th percentile deserves some comment.

RA: yes. We have emphasized this. We have also included a discussion on this in Sect. 4 (as it was also commented on by Referee #1) and a new bullet in the conclusions.

l23-26. This sounds like a political statement of the authors views. I agree that NOx

control is essential for many reasons, but cite scientific papers to support your statement.

RA: we have removed the paragraph as we agree it was too political and the manuscript is long anyway.

l21-22. This statement is unclear. Which earlier studies?

RA: We have added two references as examples.

Figures

Generally, the figure quality is quite poor and should be improved. (Some of the figures look like screen-dumps of excel plots, and the various Sweden maps (e.g. S10-S11) have awful color schemes.)

RA: We will make sure to make the final figures at a better quality to allow zooming in. In the updated manuscript we have made an effort to include high-quality figures.

We use the color schemes of Fig. S7-S10, as they allow many different levels to be shown. A person prone to colorblindness may have difficulty in interpreting some of the colors, having to turn to the text for interpretation. We have tried to explain the figures as thoroughly as possible with that in mind without making the text too heavy.

Fig. 2. The $C_5H_8$ emissions are so close to zero here that the plot doesn't show anything except that the emissions are very small. These could either be presented on a separate plot, or just described in the text. Are $C_5H_8$ emissions really so small by the way? I have seen larger estimates for Europe.

RA: The $C_5H_8$ emissions are now lifted out to a separate panel (Fig 2c) to highlight the inter-annual variations and trend. The average $C_5H_8$ emissions in the model domain in the present study is 3.1 Tg year per year. Earlier model comparisons has shown that isoprene emissions in MATCH are lower than in other, similar, models. In Langner et al.

(2012b) MATCH had average isoprene emissions of 1.6 Tg per year while the EMEP model and SILAM had 3.4 and 4.1 Tg per year, respectively. That study covered a similar domain but used meteorology from a climate model. One important conclusion from that study, was that isoprene emissions are highly uncertain and variable across models. The present C5H8 emissions are within the variability of the Langner et al. (2012b) study.

Langner, J., Engardt, M., Baklanov, A., Christensen, J.H., Gauss, M., Geels, C., Hede-gaard, G.B., Nuterman, R., Simpson, D., Soares, J., Sofiev, M., Wind, P. and Zakey, A. 2012b. A multi-model study of impacts of climate change on surface ozone in Europe. Atmos. Chem. Phys. 12, 10423-10440. doi:10.5194/acp-12-10423-2012

Fig. 3. Given the frequent discussion of the topographic location of these sites, I think a Table with altitude would also help.

RA: As we have removed a lot of the discussion on local topographical effects we do not see the need to include the altitude of the sites are needed anymore.

Fig. 4. I found the color choice unusual. Usually one uses red to indicate a warning, e.g. that data-quality is poor. Here red is used to indicate good data-quality,

RA: The colors are chosen to i/ avoid troubles for the colorblind, ii/ show well on the map in Fig. 3. The colors are coordinated in Fig. S2-S3.

Fig. 5. As noted above, I don't know what 'ozone percentiles' means if one doesn't specify which percentile. The blue and green colors here can also be hard to distin-guish.

RA: Our aim with the figure is not for the reader to understand specifically which cir-cle belongs to which percentile, but the engaged reader derive this by combining the panels with the tabulated trends in percentiles of the LONGTERM reanalysis in Sup-plementary Table S3 or extract the values from Fig. 10. We have clarified the percentile levels included in the legend.

Fig. 7. Poor quality.

RA: we have made updated the figures at a higher resolution and will make sure the final figures are zoomable.

Fig. 8. Increase the font-size for the percentile labels - they are really hard to see.

RA: we have now amended the plots including the use of larger and clearer symbols and bigger font size to increase legibility.

Fig. 9. Improve quality. I really liked the content of this Figure, and also Fig. 10, but they both look like screen dumps.

RA: The figures are updated at a higher resolution and we will make sure the final figures are zoomable.

References

Jenkin, ME, Trends in ozone concentration distributions in the UK since 1990: Local, regional and global influences, Atmos. Environ., 42, 5434-5445, 2008

Fiore, AM., Levy II, H. & Jaffe, D., A. North American isoprene influence on intercontinental ozone pollution, Atmos. Chem. Physics, 11, 1697-1710, 2011

Fiore, A., Dentener, F., Wild, O., et al., A., Multi-model estimates of intercontinental source-receptor relationships for ozone pollution, J. Geophys. Res., 114, 2009

Please also note the supplement to this comment:
https://www.atmos-chem-phys-discuss.net/acp-2017-338/acp-2017-338-AC2-supplement.zip
* * *